# Bioconcentration of Essential and Nonessential Elements in Black Sea Turbot (*Psetta Maxima Maeotica* Linnaeus, 1758) in Relation to Fish Gender

**Ira-Adeline Simionov** [1,†], **Victor Cristea** [1,2], **Stefan-Mihai Petrea** [2,†], **Alina Mogodan** [2], **Mircea Nicoara** [3], **Emanuel Stefan Baltag** [4], **Stefan-Adrian Strungaru** [5,†] and **Caterina Faggio** [6,*]

1 Multidisciplinary Research Platform (ReForm) University of Galați, 800008 Galați, Romania; ira.simionov@gmail.com (I.-A.S.); victor.cristea@ugal.ro (V.C.)
2 Food Science and Engineering Faculty, University of Galați, 800008 Galați, Romania; stefan.petrea@ugal.ro (S.-M.P.); alina.antache@ugal.ro (A.M.)
3 Faculty of Biology, Department of Biology, University of Iasi, 700505 Iasi, Romania; mirmag@uaic.ro
4 Marine Biological Station, University of Iasi, 907018 Agigea, Constanta, Romania; baltag.emanuel@gmail.com
5 Doctoral School of Geosciences, Faculty of Geography-Geology, University of Iași, 700505 Iasi, Romania; stefan.strungaru@uaic.ro
6 Department of Chemical, Biological, Pharmaceutical and Environmental Sciences, University of Messina-Italy, 98166 Sicily, Italy
\* Correspondence: cfaggio@unime.it; Tel.: +39-090-676-5213
† These authors contributed equally to this work.

**Abstract:** This study investigates the influence of gender in the bioconcentration of essential and nonessential elements in different parts of Black Sea turbot (*Psetta maxima maeotica*) body, from an area considered under high anthropogenic pressure (the Constanta City Black Sea Coastal Area in Romania). A number of 13 elements (Ca, Mg, Na, K, Fe, Zn, Mn, Cu, Ni, Cr, As, Pb and Cd) were measured in various sample types: muscle, stomach, stomach content, intestine, intestine content, gonads, liver, spleen, gills and caudal fin. Turbot adults (4–5 years old) were separated, according to their gender, into two groups (20 males, 20 females, respectively), and a high total number of samples (1200 from both groups) were prepared and analyzed, in triplicate, with Flame Atomic Absorption Spectrometry and High-Resolution Continuum Source Atomic Absorption Spectrometry with Graphite Furnace techniques. The results were statistically analyzed in order to emphasize the bioconcentration of the determined elements in different tissues of wild turbot males vs. females, and also to contribute to an upgraded characterization of the Romanian Black Sea Coast, around Constanta City, in terms of heavy metals pollution. The essential elements Mg and Zn have different roles in the gonads of males and females, as they were the only elements with completely different patterns between the analyzed groups of specimens. The concentrations of studied elements in muscle were not similar with the data provided by literature, suggesting that chemistry of the habitat and food plays a major role in the availability of the metals in the body of analyzed fish species. The gender influenced the bioaccumulation process of all analyzed elements in most tissues since turbot male specimens accumulated higher concentration of metals compared to females. The highest bioaccumulation capacity in terms of Ca, Mg, Na, Ni, As, Zn and Cd was registered in caudal fin, liver and intestine tissues. Also, other elements such as K, Fe, Cu and Mn had the highest bioaccumulation in their muscle, spleen, liver and gills tissues. The concentrations of toxic metals in Black Sea turbot from this study were lower in the muscle samples compared with the studies conducted in Turkey, suggesting that the anthropogenic activity in the studied area did not pose a major impact upon the habitat contamination.

**Keywords:** bioconcentration; biological functions; heavy metals; Black Sea turbot

## 1. Introduction

Essential metals have a role in biological functions, but in higher concentrations they may become toxic by disrupting metabolic activities [1,2]. On the other hand, nonessential metals, such as toxic metals resulting in the water from fine suspended solids, may alter the feeding rate of fish and determine a reduction in the metabolic efficiency [3,4]. Sublethal effects manifested when fish are exposed to acute metals concentrations include the damage of sensory organs and receptors, which leads to impairment of sensorial perception (namely olfaction) [4]. According to Gati et al. (2016), sediment samples with a higher value than four of the probable effect concentration quotient (PEC-Q) are more likely to have acute or toxic effects on benthic organisms [3]. In this way, integrated measures for toxicological risk ranking in fish communities were developed [5]. Accumulation of toxic metals in the environment leads to concerns, manifested by governments and scientific communities, due to potential problems associated with health risks and food poisoning [6–14]. Toxic metals can damage the liver, kidney [15], digestive system and the brain, and can produce cancer to higher rank consumers [16] in communities with preponderant fish-based diets. Furthermore, toxic metals can reach the cell nucleus and cause mutagenesis [17]. Contamination of the marine environment has risen in the last years, and is correlated with exponential human population increase and therefore, with an upward trend in anthropogenic pressure, a fact that may harm all the aquatic ecosystems [2,6,18–21]. With the exception of the organically-bound elements (hydrogen, carbon, nitrogen and oxygen), there are approximately 20 inorganic mineral elements which are considered to be essential to sustain animal life, including fish [1,2]. The physiological role of essential elements such as K, P, Na, Mg and Ca is related to skeletal structure, the maintenance of the colloidal system and regulation of the acid–base equilibrium [22]. Their concentrations in fish tissue are influenced by different factors, such as species ecology, feeding behavior, environmental variations and conditions [23]. However, the marine food webs, habitat and geographical origin are important factors that influence pollutants excretion or bioaccumulation processes in aquatic organisms [24].

As the Black Sea is considered a semi-enclosed sea and has a positive freshwater balance mainly due to high river inputs, it is exposed to heavy metals' contamination, especially in coastal areas. Fish are considered top consumers in all aquatic systems, and therefore they are continuously exposed to contaminants, a situation which involves tissues' bioaccumulation, especially in the case of demersal species, of both essential and non-essential heavy metals. Thus, as elements have the tendency to accumulate in specific hot spots, the use of demersal fish species as bio-indicators is recommended, in order to evaluate heavy metals' pollution. Thus, analyzing different fish tissues ensures scanning of this aquatic environment, considering fish mobility and capacity to continuously accumulate xenobiotics.

A series of demersal fish species as turbot (*Psetta maxima maeotica*) and whiting (*Merlangius merlangus*) can be used as bioindicators in order to evaluate the heavy metals' pollution in certain coastal areas, since they are species that do not migrate on long, transboundary distances [25]. Also, these species are involved only in local migrations within the coastal areas for spawning, feeding and wintering [25].

Therefore, the analysis of demersal fish catches in terms of heavy metals concentrations can represent an important instrument in order to evaluate the coastal areas' pollution, especially after the wintering period, when fish stock migrate closer to shore, at water depths between 20–40 m, for spawning.

Among Black Sea demersal fish species, turbot presents interest in research for the biomonitoring of polychlorinated biphenyls [26,27]; food science preservation—freshness assessment of turbot by using different biochemical and proteomics methods [28]; molecular genetics [29]; ecology and

behavior [30]; biomarker in oil pollution [31]; nitrite toxicity [32,33]; aquaculture [34]; heavy metal biomonitoring [35–40].

Several research papers conducted by Ergönül and Altindağ, 2014 [40]; Nisbet et al., 2010 [39]; Das et al., 2009 [38]; Bat et al., 2006 [37]; Tuzen 2003, 2009 [26,36] studied the concentration of potentially toxic metals and metalloids (Zn, Fe, Cu, Ni, Mn, Cr, Cd, Pb, As) in the Black Sea turbot muscle tissue captured in Turkish territorial marine waters. Other scientific articles realized by Manthey-Karl et al., 2016 [41], Martinez et al., 2010 [23] and Lourenco et al., 2012 [22] studied the concentration of essential macroelements (Ca, Mg, Na, K), microelements (Zn, Fe, Cu, Ni, Mn) and potentially toxic metals (Cd, Pb, As) in the muscle tissue of wild turbot captured in the Atlantic Ocean, respectively, in farmed turbot from Portugal. However, the aforementioned studies which focused on essential, non-essential elements and toxic metals in the Black Sea turbot organism are incomplete, since only potential toxic microelements and trace elements were determined. The macroelements are important to be determined in order to identify their capacity to influence micro- and trace elements' bioaccumulation. Also, this type of study can contribute to a better evaluation of certain Black Sea coastal areas, considered under a continuous anthropogenic pressure.

This study investigates the influence of turbot gender, caught on the Romanian Black Sea Coast, in bioconcentrations of both essential (Ca, Cu, Fe, K, Mg, Mn, Na, Zn) and nonessential elements (As, Cd, Ni) in various tissues (gills: Gi, stomach: St, intestine: In, liver: Lv, spleen: Sp, gonads: Go, muscles: Mu, caudal fin: Cf), including stomach and intestine content. This evaluation will also contribute to an upgraded characterization of the Romanian Black Sea Coast, in terms of heavy metals' pollution.

## 2. Materials and Methods

### 2.1. Sampling Area Description

The Black Sea is a large and deep, meromictic environment connected with the global ocean via a system of straits and intermediate seas [42]. The sea is located in the East–West depression between two alpine fold belts, with the Pontic Mountains to the South and Caucasus mountains to the Northeast [43]. It is considered to be an isolated sea from the World Ocean that is under anthropogenic pressure, particularly in the shore area [43]. Six riparian countries are neighboring the Black Sea: Romania, Bulgaria, Turkey, Georgia, Russian Federation, and the Ukraine [26]. The Black Sea was characterized as a fertile sea in the early 1990s, with the most productive areas located in the Northwest and Northeast, due to the affluent rivers rich in nutrients [25]. The sea surface area is 423,000 km$^2$, and the maximum depth is around 2200 m [44]. The Black Sea annually receives a large amount of freshwater input from the Danube, Dniepr and Dniester rivers, consisting of 350 km$^3$ [43]. The resident human population located along the coastline is approximately 16 million people, with another 4 million visitors during the touristic season [45].

This area of study was chosen, since it is the most intensely populated from the Romanian coastline, and also, the main anthropogenic activities (tourism, municipal wastewater effluents, industry, cargo shipping and oil refinery) are concentrated here [43,46]. The map of the Romanian Black Sea targeted coastal area is presented in Figure 1.

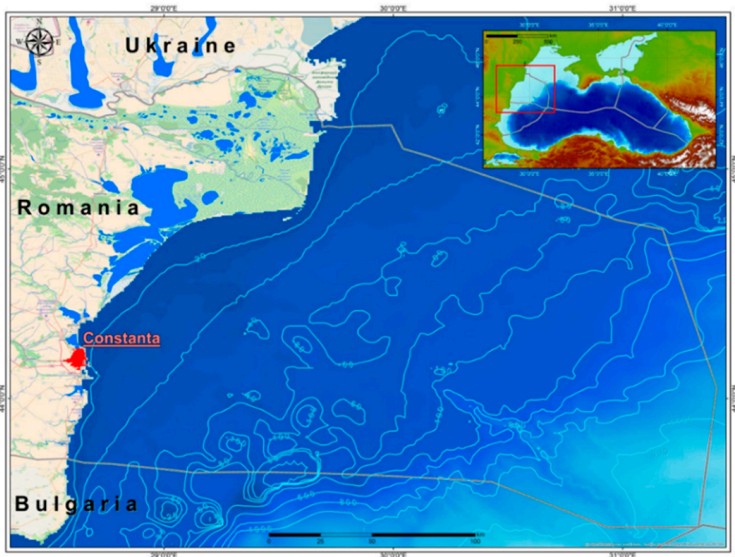

**Figure 1.** Romanian Black Sea targeted fishing territorial area and Constanta City fish market location.

*2.2. Sampled Biological Material*

Since fish stocks are affected by high anthropogenic pressure, the biological material in the present study is represented by a number of 40 Black Sea turbot specimens. The biological material was selected from a larger number of specimens purchased fresh from the fish market located in Constanta City and its surrounding area, in April 2016 (Figure 1). All turbot specimens were caught in the fishing areas along the coastline of Constanta City, by using bottom (turbot) gill nets with a minimum mesh size of 180 mm.

Flatfish are considered to be suitable bioindicators due to their ecological importance, sensitivity and close contact to sediment substrates [47]. In particular, flatfish have been widely pointed out in ecotoxicological studies carried out in estuarine, coastal and marine environments, due to their benthic behavior [48]. Turbot is a marine demersal teleost flatfish which is distributed from the Atlantic coast of Europe, to the Baltic, Black Sea and Mediterranean Sea [29,34]. *Psetta maxima maeotica*, known as The Black Sea turbot, is an endemic subspecies with high economic value [34,49]. It has distinct bony tubercles developed on both sides of the dorsal part, which are larger than the eyes [50]. The adults mainly feed on demersal (whiting, red mullet and gobies) and pelagic (anchovy, sprat, horse mackerel, shad) small-sized fish [26]. The diet also includes crustaceans (shrimps, crabs), mollusks and Polychaeta worms [25].

*2.3. Sample Preparation*

The studied material was taxonomically identified as *Psetta maxima maeotica* (Linnaeus, 1758), by using both an online fish taxonomic identification database [51] and a taxonomic identifier for Black Sea fish species [52]. The gender identification was made in situ since we had the permission to do a small abdominal incision, with plastic laboratory instruments, for each fish, to determine the sex gender, in order assure the targeted sampling number for this study (20 males and 20 females). This was also useful for the identification and selection of the fish specimens that had stomach content. A number of 40 turbot specimens (20 males and 20 females), were placed in polyethylene bags, stored on ice and transported to the laboratory, where they were frozen at −20 °C, until the samples were prepared and analyzed. Before the analysis, the specimens were dissected by using plastic laboratory instruments, for avoiding samples' contamination. Each individual was biometrically measured, and the individual biomass was determined before being frozen at −20 °C, since according to different authors, the fish size can influence the bioaccumulation process [53–55].

Based on biometric measurements, all specimens were considered as part of a cohort of 4–5 year-old adults, capable for reproduction, according to Maximov et al. [56], that collected samples, in its scientific study, from a similar area, being the Black Sea Coast, Constanta City. The age of turbot specimens was determined according to the Eryilmaz and Dalyan method [57].

After dissection, the following sample types were collected (1 g each) for analysis: muscle, stomach, stomach content, intestine, intestine content, gonads, liver, spleen, gills and caudal fin. All the analyzed specimens had over 70% stomach content, reported to the entire stomach capacity. The samples were several times washed with ultrapure, deionized water, well chopped, grinded and homogenized. This procedure was applied for each specimen. A total number of 30 samples per studied fish were prepared (three replicates per sample type) for element analyses. Each sample was weighed to a mass of 1 g and introduced into a previously decontaminated TFM pressure vessel. For digestion, 4 mL of nitric acid 65% Suprapur (certified Merck Germany) and 2 mL of hydrogen peroxide EMSURE 30% (certified Merck Germany) were added [58]. The vessels were introduced in the Speedwave MWS-2 produced by Berghof, using a 3-step program for digestion [44]. After cooling, the liquid samples were transferred into 50 mL volumetric flasks filled up with ultrapure water produced by a LaboStar™3/7 TWF (Siemens) purification system from double-distilled water. The total amount of prepared and analyzed samples in this study was equal to 1200 (10 samples per each fish specimen, in triplicate).

For the metal quantification analyses two methods were used: flame atomic absorption spectrometer FL-AAS (equipment GBC Avanta, Australia), for calcium (Ca), magnesium (Mg), sodium (Na), potassium (K), iron (Fe), zinc (Zn) and high resolution continuum source atomic absorption spectrometer with graphite furnace HR-CS GF-AAS (equipment ContrAA 600-Analytik Jena, Germany), for copper (Cu), manganese (Mn), nickel (Ni), arsenic (As), cadmium (Cd), lead (Pb) and chromium (Cr). Certified reference materials (CRMs) are recognized to be an essential tool in assuring the accuracy and establishing the traceability of the results of measurements [59]. Therefore, the reference material for fish muscle (ERM-BB422 type certified reference material) was analyzed, certified by the Joint Research Center Institute for Reference Materials and Measurements. This reference material was prepared in six replicates, following the same protocol as a normal sample [44], in order to eliminate any determination errors, and the results obtained were presented in Table 1. The certificate of analysis contained the following elements: As, Cd, Cu, Fe, Hg, I, Mn, Se, Zn, Ca, Cl, K, Mg and Na.

**Table 1.** Method for assuring the accuracy and establishing the traceability of the measurements results, based on the use of certified reference materials (CRMs) ($\mu g \cdot g^{-1}$ dry weight).

| Measured Element | Certified Value Mean ± SD ($\mu g \cdot g^{-1}$) | Measured Value Mean ± SD ($\mu g \cdot g^{-1}$) | Analytical Method | Number of Replicates |
|---|---|---|---|---|
| As | 12.7 ± 0.7 | 11.3 ± 0.4 | HR-CS GF-AAS | 6 |
| Ca | 342 | 340 ± 6 | FL-AAS | 6 |
| Cd | 0.0075 ± 0.0018 | 0.0077 ± 0.002 | HR-CS GF-AAS | 6 |
| Cu | 1.67 ± 0.16 | 1.62 ± 0.27 | HR-CS GF-AAS | 6 |
| Fe | 9.4 | 9.5 ± 0.3 | FL-AAS | 6 |
| K | 21400 | 21381 ± 21 | FL-AAS | 6 |
| Mg | 1370 | 1372 ± 7 | FL-AAS | 6 |
| Mn | 0.368 ± 0.028 | 0.361 ± 0.034 | HR-CS GF-AAS | 6 |
| Na | 2800 | 2771 ± 27 | FL-AAS | 6 |
| Zn | 16 ± 1.1 | 15 ± 1.7 | FL-AAS | 6 |

*2.4. Statistical Analysis*

First, the Levene's homogeneity variance test and the Kolmogorov–Smirnov normality test were performed to study the data distribution in both experimental groups (males and females). The result of each test proved that the data had a normal distribution. After this step there was performed a one-way analysis of variance (ANOVA) for the significance of the variance for each element concentration between tissues that were sampled from both genders, followed by multiple comparisons Tukey HSD

test, in order to determine which of the means are different. The t-Test was performed to observe any significant differences between females and males regarding all the concentrations of studied elements for each sample type. All statistical analysis was carried out using OriginPro v.9.3. (2016) software (OriginLab Corporation, Northampton, Massachusetts, USA). All the data for metal concentration were presented in this study as average ± standard deviation (SD). The mean values registered for different metals' concentrations in the muscle tissue were centralized and compared to data reported in other studies, which sampled muscle tissue from aquaculture and wild turbot.

The bioconcentration factor (BCF) was calculated for dietary exposure according to formula (1) below [46]. All specimens used in this study had food in their stomach. The main route of absorption of the elements in the body was considered to be the food and sediments that were ingested, according to Bury et al., who pointed out that the diet is the main source of essential metals in the fish body [60]. The value of this factor indicates the capacity of a specific biological sample to accumulate a certain part from the total element input in the fish body.

$$BCF = \frac{\text{element concentration in biological sample } (f.w.)}{\text{element concentration in stomach content } (f.w.)}$$

## 3. Results and Discussions

The results of biometric measurements and individual biomass of the studied turbots are presented in Table 2. In this study, the males group had significant (*p < 0.05) higher values than the female group for all biometric variables according to t-test results.

**Table 2.** Biometric measurements and individual biomass of the studied fish (mean ± SD).

| Biometric Measurement | Gender | |
|---|---|---|
| | Female | Male |
| Total length (cm) | 44.60 ± 1.86 | 48.3 ± 1.55 |
| Maximum width (cm) | 32.6 ± 2.15 | 35.1 ± 0.28 |
| Individual biomass (kg) | 1.83 ± 0.29 | 2.32 ± 0.12 |
| Distance between eyes (cm) | 1.4 ± 0.05 | 1.5 ± 0.02 |
| Caudal fin length (cm) | 8.24 ± 0.05 | 10.18 ± 0.3 |
| Head length (cm) | 8.74 ± 0.67 | 10.21 ± 0.49 |
| Head maximum width (cm) | 8.9 ± 0.02 | 12.56 ± 0.56 |

The Pb and Cr were below the detection limit of quantification (LOQ) for the calibration curve method (LOQ Pb-0.032 $\mu g \cdot L^{-1}$, LOQ Cr-0.3 $\mu g \cdot L^{-1}$) in all analyzed samples, excepting for the stomach samples, where Cr was measured. Females had in average in the stomach 0.09 ± 0.001 $\mu g \cdot g^{-1}$ and males 0.10 ± 0.002 $\mu g \cdot g^{-1}$ chromium, respectively. For the rest of analyzed elements (Ca, Mg, K, Na, Zn, Fe, Cu, Mn, Ni, As and Cd), we observed significant variations (*p < 0.05, ANOVA) between the studied tissues (muscle, stomach, stomach content, intestine, intestine content, gonads, liver, spleen, gill and caudal fin) for males and females. This was shown by the multiple comparisons results of the Tukey test.

The values of the elements' concentrations in the studied samples are presented in Table 3. Significant differences (*p < 0.05*) between males and females of Black Sea turbot were observed. In gills, significant differences (*p < 0.05*) between genders were recorded for magnesium (Mg), sodium (Na), potassium (K), zinc (Zn), iron (Fe), copper (Cu), manganese (Mn), arsenic (As) and cadmium (Cd). The stomach content analysis was considered to be a relevant measurement of the metal dietary intake in the turbot body. Significant differences (*p < 0.05*) were recorded between male and female groups in terms of the concentrations of all studied elements, except Cd, in stomach content samples. This could be related to turbot diet and its mobility in various areas with different elements' concentrations, where the food resources are located [25]. Similar results were observed when analyzing the stomach

samples. The intestine is the next organ where digestion continues. In the intestinal content samples, the concentrations of Ca, and respectively Mg, Na, K, Zn, Cu, Ni and As, differ significantly, between males and females specimens. In analyzed intestinal tissue, significant differences were recorded between genders, in terms of Mg, Na, K, Zn, Fe, Cu, Ni and As concentrations. Female and male specimens of Black Sea turbot may have different physiological needs of elements, a fact revealed by the results recorded for all types of samples analyzed, where the accumulation of elements was found to be different. For instance, in gonads, females accumulated more zinc and less magnesium, compared to males.

Figure 2 emphasizes the variations and affinity of elements in the body of male and female specimens of Black Sea turbot, based on the registered results recorded from the analyzed sample tissues. It must be noted that the studied males registered superior biometric measurements and individual biomass values, compared to females. Therefore, for example, in case of gonads, it can be observed a specific accumulation trend for males, compared to females, for each analysed element, emphasizing the different role of these elements in gonads, for each gender.

The bioconcentration factor of the essential and nonessential elements in the Black Sea turbot was calculated from the dietary intake. In this case, the dietary intake is represented by the stomach content. It must be highlighted that if bioaccumulation analysis is conducted in relation to sediments, the results could be not entirely correct, since Black Sea turbot specimens are mobile benthic organisms. The chemistry of Black Sea sediments differs from one area to another [46], and the location where the captured specimens lived cannot be identified exactly, since turbot is capable of traveling significant distances in search for food [25]. Thus, in the present study it has been chosen to calculate the BCF of non-essential and essential elements, for all analyzed tissues, in relation to stomach content samples. In Table 4 there are presented the values of the bioconcentration factor calculated from dietary exposure, both for males and females specimens. Each element manifests a bioconcentration propensity for a specific organ. The capacity trend of bioconcentration was the same for males and females, excepting the gonads, where statistically different results, ($p < 0.05$) from ANOVA, followed by the Tukey HSD test, were recorded. The exposure history for each specimen and the area from which they came from are both unknown.

The role of essential and nonessential elements on various fish species was described by scientific papers [61,62]. Not all of the elements were able to be studied, a situation that generates incomplete answers. Our results were compared with studies regarding the essential and nonessential elements from muscle samples of farmed turbot [22] or wild turbot from the Atlantic Ocean [23,41]. The scientific studies on turbot were driven by the fact that this specie has a high economic value. Thus, not all the elements were investigated in detail. Table 5 presents a comparative study between mean concentrations of elements recorded in this present study and the concentrations reported, in their research, by other authors. The comparative study revealed a high variability related to essential and non-essential elements' concentrations in turbot specimens from the Black Sea and other turbot subspecies, from different parts of the world. So far, this study covers the lack of published data in the present area of interest, between the years 2006–2016.

The deficiency of essential elements in the fish body, such as Na, K, P, Ca, Mg, Mn, Fe, Cu and Zn, may lead to improper or poor enzymes' metabolic functions, which will cause organ malfunctions, chronic diseases and finally, death [22]. As metals have different affinity for organs in the fish body, a fact explained by the metabolic role of each metal [63], trace and ultra-trace elements as Zn, Fe, Cu, Mn, Cr and Ni are essential elements and important components of hormones and enzymes, and also, for enzyme activation.

They are associated with specific proteins in metalloenzymes and with special roles in catalytic functions [22]. Toxic elements such as Cd and Pb can be harmful if ingested for a long-time period. Even essential trace elements such as Zn, Cu and Cr can produce toxic effects if the intake is excessively high [23].

For instance, calcium and potassium play an important role in the skeletal structure of aquatic organisms, therefore fish and other aquatic organisms can absorb Ca and K from water, and their Ca requirement is covered by the capacity to absorb this element directly from water [61]. The Ca deficiency was not observed in the samples from the present study (Table 3). However, the Ca concentration in the muscle tissue, recorded in present study, is slightly higher compared to mean Ca concentration registered by Lourenco et al. [22], from turbot muscle. This may be explained by the differences of diets and ecosystems, since the biological material from the aforementioned study [22] was reared in aquaculture production systems, and therefore in controlled conditions.

The K concentration from muscle tissue registered the highest values, compared to the rest of the studied elements, and it was almost twice as high as the K concentration in the turbot muscle reported by Lourenco et al. [22]. Since diet is the main source of K in fish, the difference between farmed turbot and wild turbot in terms of K concentration in muscle tissues is justified, a situation similar also for Ca. Once absorbed, K accumulates in soft tissues such as liver, kidney and muscle [61]. It can be observed (Figure 2) that the K pathway shows the same trend in both genders, with the highest concentrations in muscle and liver tissues.

In the fish body, Mg has an important role in transport, a fact that implies gills, intestinal and renal transport [64]. The mean concentration in samples analyzed in this present study was twice higher than those measured by Lourenco et al. [22] and Manthey-Karl et al. [41]. In fish, Mg is found mineralized in bony tissues and the hard tissues of the dermal and skeletal bones [64]. The pathway of Mg in the analyzed turbot samples indicates differences between both groups (males and females) in intestine and intestine content, where higher concentrations were recorded among the female group, compared to the male group. However, the concentration of Mg in the rest of the analyzed tissues follows the same trend for both groups, with the highest values registered in the caudal fin.

**Table 3.** Elements' concentration (mean ± standard deviation (SD)) in analyzed samples, reported as $\mu g \cdot g^{-1}$ fresh weight. The values marked with letter "a" are significantly different (* $p < 0.05$ for *t*-Test) between the studied females and males.

| Element | Gender | Gills | Stomach | Stomach Content | Intestine | Intestine Content | Liver | Spleen | Gonads | Muscles | Caudal Fin |
|---|---|---|---|---|---|---|---|---|---|---|---|
| Ca | Female | 180.2 ± 1.1 | 208.9 ± 24.9[a] | 347.3 ± 50.5[a] | 307.6 ± 31.2 | 537.9 ± 29.6[a] | 97.8 ± 3.5[a] | 143.4 ± 5.6[a] | 86.2 ± 16.7[a] | 175.3 ± 13.7[a] | 4127.1 ± 51 |
|  | Male | 179 ± 5.9 | 122.6 ± 0.5[a] | 197.6 ± 19.2[a] | 274.1 ± 75.9 | 425.9 ± 46.4[a] | 113.8 ± 6.4[a] | 113.4 ± 14.4[a] | 105.7 ± 11.8[a] | 278.3 ± 98.6[a] | 4090.9 ± 77.6 |
| Mg | Female | 149.9 ± 12.4[a] | 498.6 ± 22.1[a] | 546.7 ± 8.6[a] | 977.7 ± 31.6[a] | 1285.1 ± 170[a] | 524.3 ± 37.5[a] | 596.2 ± 54.8[a] | 228.5 ± 20.3[a] | 482.5 ± 28.9[a] | 1574.2 ± 75.3[a] |
|  | Male | 273.1 ± 29.9[a] | 372.1 ± 49.5[a] | 489.4 ± 45.1[a] | 793.5 ± 58.5[a] | 443.1 ± 374.3[a] | 345.2 ± 6.3[a] | 415.1 ± 30.9[a] | 492.8 ± 34.1[a] | 553.5 ± 32.9[a] | 1468.4 ± 9.03[a] |
| Na | Female | 1029.1 ± 348.8[a] | 1956.3 ± 381.1[a] | 2069.2 ± 179.7[a] | 1592.4 ± 149.5[a] | 2035.2 ± 19.2[a] | 1375.5 ± 85.2[a] | 1868.9 ± 152.1 | 1599.5 ± 89.2[a] | 926.5 ± 69.1[a] | 2824.5 ± 168.7[a] |
|  | Male | 2713.8 ± 125.4[a] | 2456.4 ± 52.8[a] | 2750.7 ± 142.2[a] | 2269.1 ± 137.7[a] | 2179.4 ± 147.3[a] | 1647.5 ± 12.9[a] | 1888.3 ± 94.4 | 2356.7 ± 19.8[a] | 1306.5 ± 56.7[a] | 3295.9 ± 300.8[a] |
| K | Female | 986.2 ± 36.5[a] | 1971.5 ± 575[a] | 2273.4 ± 85[a] | 2585.9 ± 76.1[a] | 3330.5 ± 279.5[a] | 5516.1 ± 47.7[a] | 4899.1 ± 162.6[a] | 3011 ± 154.6[a] | 5917.3 ± 165.7[a] | 1162.7 ± 41.4[a] |
|  | Male | 1933.7 ± 453[a] | 3410.3 ± 59.7[a] | 3883.5 ± 283.9[a] | 4576.6 ± 286.9[a] | 4204.1 ± 91.2[a] | 4261.9 ± 570.2[a] | 4678.1 ± 353.5[a] | 3352.6 ± 122.4[a] | 6085 ± 284.35[a] | 1415.7 ± 117.9[a] |
| Zn | Female | 22.1 ± 1.1[a] | 23.2 ± 1.9[a] | 17.9 ± 1.1[a] | 23.93 ± 0.55[a] | 25.6 ± 4.7[a] | 26.5 ± 0.8[a] | 36 ± 0.56[a] | 39.1 ± 2.6[a] | 10.3 ± 2.9[a] | 34.2 ± 1.3[a] |
|  | Male | 24 ± 1.7[a] | 34.8 ± 2.2[a] | 38.6 ± 1.06[a] | 31.4 ± 2.3[a] | 40.3 ± 7.3[a] | 30.6 ± 1.7[a] | 32.3 ± 0.2[a] | 31.6 ± 0.3[a] | 14.06 ± 1.43[a] | 47.15 ± 0.07[a] |
| Fe | Female | 144.8 ± 18.2[a] | 25.5 ± 5.1[a] | 56.1 ± 23.03[a] | 20.3 ± 3.7[a] | 21.05 ± 0.55 | 51.7 ± 5.9[a] | 169.1 ± 19.3[a] | 15.1 ± 1.8[a] | 12.25 ± 2.39 | 52.5 ± 18.4 |
|  | Male | 109.4 ± 6.5[a] | 20.3 ± 4.2[a] | 30.05 ± 3.02[a] | 28.5 ± 2.1[a] | 23.25 ± 4.7 | 68.9 ± 6.5[a] | 154.6 ± 9.8[a] | 10.4 ± 0.1[a] | 6.01 ± 1.25 | 25.09 ± 7.68 |
| Cu | Female | 0.47 ± 0.01[a] | 0.5 ± 0.03[a] | 0.72 ± 0.05[a] | 0.71 ± 0.02[a] | 1.06 ± 0.14[a] | 2.7 ± 0.1[a] | 0.62 ± 0.04[a] | 0.7 ± 0.07[a] | 0.15 ± 0.01 | 0.58 ± 0.09[a] |
|  | Male | 0.61 ± 0.03[a] | 0.99 ± 0.03[a] | 1.17 ± 0.06[a] | 1.29 ± 0.09[a] | 1.52 ± 0.11[a] | 3.4 ± 0.3[a] | 1.04 ± 0.18[a] | 0.62 ± 0.01[a] | 0.15 ± 0.01 | 0.72 ± 0.001[a] |
| Ni | Female | 0.13 ± 0.02 | 0.10 ± 0.001[a] | 0.09 ± 0.00[a] | 0.08 ± 0.001[a] | 0.10 ± 0.001[a] | 0.2 ± 0.001[a] | 0.16 ± 0.02[a] | 0.08 ± 0.01 | 0.09 ± 0.02[a] | 2.15 ± 0.66[a] |
|  | Male | 0.12 ± 0.01 | 0.13 ± 0.001[a] | 0.07 ± 0.001[a] | 0.12 ± 0.001[a] | 0.12 ± 0.001[a] | 0.14 ± 0.001[a] | 0.10 ± 0.001[a] | 0.07 ± 0.00 | 0.12 ± 0.02[a] | 0.99 ± 0.07[a] |
| Mn | Female | 1.37 ± 0.06[a] | 0.22 ± 0.11[a] | 1.05 ± 0.09[a] | 0.52 ± 0.04 | 1.1 ± 0.2 | 0.6 ± 0.3[a] | 0.38 ± 0.03 | 0.16 ± 0.07 | 0.19 ± 0.01 | 0.04 ± 0.001 |
|  | Male | 1.66 ± 0.01[a] | 0.42 ± 0.03[a] | 0.30 ± 0.17[a] | 0.56 ± 0.07 | 1.05 ± 0.02 | 0.5 ± 0.15[a] | 0.30 ± 0.01[a] | 0.21 ± 0.11 | 0.15 ± 0.07[a] | 0.05 ± 0.01[a] |
| As | Female | 0.18 ± 0.03[a] | 0.73 ± 0.02[a] | 0.85 ± 0.16[a] | 1.8 ± 0.4[a] | 2.06 ± 0.31[a] | 9.31 ± 4.88 | 0.70 ± 0.68[a] | 0.18 ± 0.04[a] | 3.82 ± 0.95[a] | 9.87 ± 1.34[a] |
|  | Male | 0.21 ± 0.01[a] | 1.90 ± 0.49[a] | 1.47 ± 0.28[a] | 2.3 ± 0.5[a] | 2.85 ± 0.04[a] | 7.52 ± 2.43 | 1.42 ± 0.7[a] | 0.88 ± 0.12[a] | 3.80 ± 1.32 | 12.31 ± 0.89[a] |
| Cd | Female | 0.06 ± 0.01[a] | 0.03 ± 0.001 | 0.04 ± 0.01[a] | 0.09 ± 0.02 | 0.11 ± 0.01 | 0.09 ± 0.01 | 0.05 ± 0.001[a] | 0.03 ± 0.0001 | 0.03 ± 0.001 | 0.03 ± 0.001[a] |
|  | Male | 0.08 ± 0.01[a] | 0.03 ± 0.001 | 0.08 ± 0.01[a] | 0.11 ± 0.03 | 0.11 ± 0.05 | 0.09 ± 0.02 | 0.03 ± 0.001[a] | 0.03 ± 0.001 | 0.03 ± 0.001 | 0.13 ± 0.05[a] |

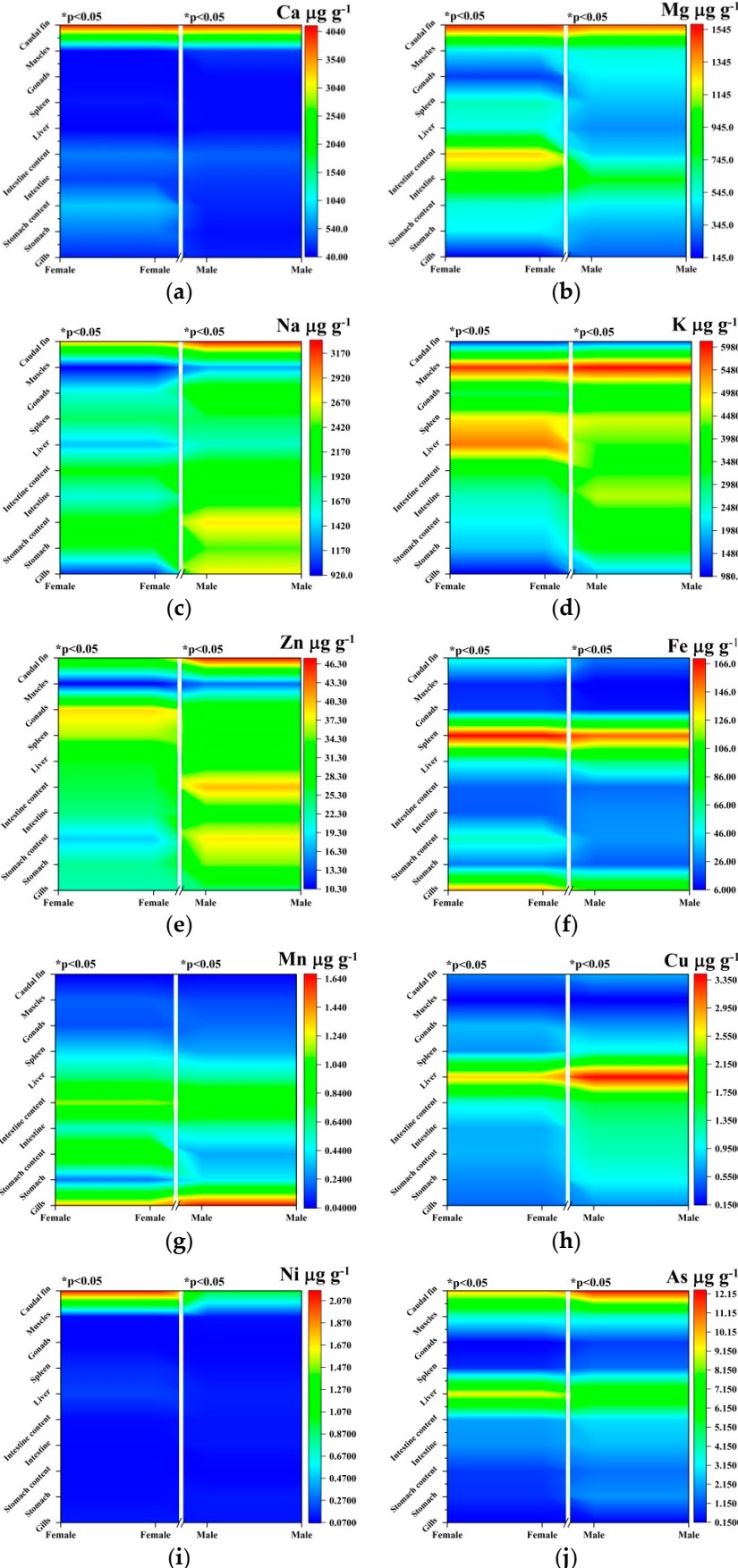

**Figure 2.** *Cont.*

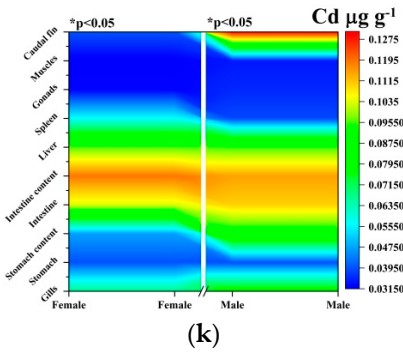

**(k)**

**Figure 2.** The representation of elements' affinity and variation in the body of male and female specimens of Black Sea turbot, based on the registered results ($\mu$g·g$^{-1}$ fresh weight). (one-way analysis of variance (ANOVA) was applied for males and females in order to determine the variation of each studied element; *$p < 0.05$). (**a**) The Ca concentration in analysed turbot tissues (**b**) The Mg concentration in analysed turbot tissues (**c**) The Na concentration in analysed turbot tissues (**d**) The K concentration in analysed turbot tissues (**e**) The Zn concentration in analysed turbot tissues (**f**) The Fe concentration in analysed turbot tissues (**g**) The Mn concentration in analysed turbot tissues (**h**) The Cu concentration in analysed turbot tissues (**i**) The Ni concentration in analysed turbot tissues (**j**) The As concentration in analysed turbot tissues (**k**) The Cd concentration in analysed turbot tissues.

**Table 4.** The bioconcentration factor calculated from dietary exposure in Black Sea turbot (males and females).

| Element | Male | | | | | | | | |
|---|---|---|---|---|---|---|---|---|---|
| | Stomach | Intestine Content | Intestine | Liver | Spleen | Gonads | Muscle | Caudal Fin | Gills |
| Ca | 0.62 | 2.16 | 1.39 | 0.58 | 0.57 | 0.54 | 1.41 | 20.70 | 0.91 |
| Mg | 0.76 | 0.91 | 1.62 | 0.71 | 0.85 | 1.01 | 1.13 | 3.00 | 0.56 |
| Na | 0.89 | 0.79 | 0.82 | 0.60 | 0.69 | 0.86 | 0.47 | 1.20 | 0.99 |
| K | 0.88 | 1.08 | 1.18 | 1.10 | 1.20 | 0.86 | 1.57 | 0.36 | 0.50 |
| Zn | 0.90 | 1.05 | 0.81 | 0.80 | 0.84 | 0.82 | 0.36 | 1.22 | 0.62 |
| Fe | 0.68 | 0.77 | 0.95 | 2.29 | 5.14 | 0.35 | 0.20 | 0.83 | 3.64 |
| Cu | 0.85 | 1.29 | 1.10 | 2.91 | 0.89 | 0.53 | 0.13 | 0.62 | 0.52 |
| Mn | 1.39 | 3.43 | 1.82 | 1.89 | 1.00 | 0.69 | 0.50 | 0.17 | 5.41 |
| Ni | 1.69 | 1.61 | 1.59 | 1.81 | 1.35 | 0.98 | 1.55 | 12.86 | 1.58 |
| Cd | 0.44 | 1.28 | 1.23 | 1.06 | 0.43 | 0.40 | 0.38 | 1.45 | 0.90 |
| As | 1.29 | 1.94 | 1.63 | 5.11 | 0.97 | 0.60 | 2.58 | 8.35 | 0.15 |
| | **Female** | | | | | | | | |
| Ca | 0.59 | 1.54 | 0.88 | 0.28 | 0.41 | 0.24 | 0.5 | 11.89 | 0.42 |
| Mg | 0.91 | 2.35 | 1.79 | 0.96 | 1.09 | 0.42 | 0.88 | 2.88 | 0.27 |
| Na | 0.95 | 0.98 | 0.77 | 0.66 | 0.90 | 0.77 | 0.45 | 1.36 | 0.50 |
| K | 0.87 | 1.46 | 1.14 | 2.43 | 2.15 | 1.32 | 2.60 | 0.51 | 0.43 |
| Zn | 1.30 | 1.43 | 1.33 | 1.48 | 2.01 | 2.18 | 0.57 | 1.91 | 1.24 |
| Fe | 0.46 | 0.38 | 0.36 | 0.92 | 3.01 | 0.27 | 0.22 | 0.94 | 2.58 |
| Cu | 0.82 | 1.47 | 0.98 | 3.84 | 0.86 | 1.04 | 0.22 | 0.80 | 0.66 |
| Mn | 0.21 | 1.09 | 0.50 | 0.62 | 0.37 | 0.16 | 0.18 | 0.04 | 1.30 |
| Ni | 1.07 | 1.14 | 0.96 | 2.16 | 1.74 | 0.89 | 1.04 | 23.15 | 1.40 |
| Cd | 0.87 | 2.46 | 2.16 | 2.05 | 1.25 | 0.71 | 0.71 | 0.86 | 1.48 |
| As | 0.86 | 2.42 | 2.20 | 10.92 | 0.82 | 0.22 | 4.48 | 11.57 | 0.22 |

The Na is known to be involved in important physiological activities, such as sodium channels, and in the osmotic pressure regulation [65]. The blood of a marine fish contains salt at a concentration of one-third of that of full-strength seawater, thus, fish drink a specific amount of water (0.5% body weight/hour) to regulate the osmotic pressure [66]. The mean Na values in the turbot muscles are similar with the results obtained by Manthey-Karl et al. [41], explained by the fact that both studies (present and aforementioned) analyzed samples from wild turbot specimens.

The Na pathway in the body of analyzed turbot specimens did not significantly differ between genders, even if males concentrated more Na in the caudal fin, stomach content and gills, compare to females. The main route of NaCl uptake is the intestine, while gills represent the main excretion route [66].

Zinc mean concentrations in our turbot muscles' samples were similar to the results obtained by Ergönül and Altindağ [40] for wild turbot from the Black Sea. Twice higher Zn concentrations were registered in turbot muscle, compared to the results recorded by Lourenco et al. [22] and Manthey-Karl et al. [41]. Also, the Zn values of Bat et al. [37] and Tuzen [36] (which also analyzed Black Sea turbot specimens) were two and three times higher, respectively, in the turbot muscles, compared to the concentrations registered in the present study. This may be due to the applied analysis methodology (the aforementioned studies did not use CRM for assuring the accuracy and establishing the traceability of the measurements' results) or sampling area (the Black Sea Turkish coast has been observed to be more polluted in terms of heavy metals, compared to the Romanian coast [67]). Zn is an essential micro-nutrient, which acts as a cofactor for numerous enzymes and redox proteins, such as DNA and RNA polymerase, carbonic anhydrase or alkaline phosphatase [68]. The Zn pathway in the body of analyzed turbot specimens differ for each of the two groups (males and females). This element is involved in fish development and reproduction. In Figure 2 it can be observed that females concentrated most Zn in the gonads, while males concentrated Zn mostly in the caudal fins.

The mean iron concentration in the muscle tissue, registered in the present study, is three times higher compared to the concentrations recorded by Lourenco et al. [22], but at the same time, lower compared to other studies [36,37,39,40]. According to Lourenco et al. [22], Fe was present in concentrations of approximately 5.0 $\mu g \cdot g^{-1}$ in marine fish species, which was similar to the present study results. The Fe tends to accumulate in hepatic tissue due to the physiological role of the liver in the synthesis of blood cells and hemoglobin [69]. The Figure 2 reveals that the pathway of Fe in turbot bodies is similar between males and females, with the highest values in the spleen, which is known as the main destruction site of aged red cells.

The Cu mean concentrations in turbot muscles were similar to the results of Lourenco et al. [22] and Martinez et al. [23], but compared with other studies [37,39], the values recorded in the present study were fourteen, respectively, thirty-three times lower. The Cu pathways in the body of turbot specimens from both analyzed groups are similar, with the highest concentrations recorded in liver tissue.

The mean of Ni concentration in turbot muscles was similar to the values found by Martinez et al. [23], but more than thirty times lower than the concentrations reported by Nisbet et al. [39] and Bat et al. [37] for the Black Sea turbot meat. The pathways of Ni in the turbot body are similar both in male and female analyzed specimens, with the highest concentrations recorded in caudal fins.

The Mn mean concentration in the turbot muscles is similar to the values obtained by Martinez et al. [23]. The Mn is an essential micronutrient, and it is a component of metalloenzymes and an activator for other enzymes. The pathway of Mn in the turbot body follows the same trend in male and female analyzed specimens, with the highest concentration recorded in gills tissue.

The mean As concentration recorded in turbot muscles is similar to the results of Manthey-Karl et al. [41]. The pathway of As in the body of turbot analyzed specimens was similar for both genders, with the highest concentration in the caudal fin and liver. According to Manthey-Karl et al. [41], the seafood can have naturally high levels of As.

The Cd is a genotoxic metal for fish [70] that has already been classified as a carcinogen in humans, and is considered toxic to aquatic organisms by inducing the oxidative stress and immunotoxicity. The level of Cd in water can reach 0.05 $mg \cdot L^{-1}$ [71]. The mean Cd concentration in the muscle of turbot is similar to the results reported by Ergönül and Altindağ [40], Nisbet et al. [39] and Bat et al. [37]. The values recorded by Tuzen [36] for Cd concentration in the muscle of Black Sea turbot were three times higher, compared to the concentrations recorded in this present study. The pathway of Cd in turbot body is similar for both males and females, except caudal fin, where males concentrated more Cd than the females.

**Table 5.** Essential and non-essential elements concentrations in the turbot muscle (mean ± SD) µg·g$^{-1}$, f.w.: fresh weight, d.w.: dry weight.

| References | Macroelements | | | | Microelements | | | | | | | | |
| --- | --- | --- | --- | --- | --- | --- | --- | --- | --- | --- | --- | --- | --- |
| | Ca | Mg | Na | K | Zn | Fe | Cu | Ni | Mn | As | Cd | Cr | Pb |
| Present study (f.w.) | 176.8 ± 122.5 | 518.1 ± 46.6 | 1116.5 ± 199.7 | 6001.2 ± 241.8 | 12.18 ± 2.95 | 9.13 ± 3.63 | 0.15 ± 0.01 | 0.10 ± 0.02 | 0.17 ± 0.05 | 3.81 ± 1.12 | 0.03 ± 0.001 | < LOD | < LOD |
| [41] (f.w.) | 90 ± 17 | 240 ± 20 | 1084 ± 96 | 2836 ± 183 | 6.1 ± 1.2 | - | - | - | - | 4.6 ± 1.10 | - | - | - |
| [40] (d.w.) | - | - | - | - | 21.4 ± 5.38 | 48.6 ± 9.06 | 0.75 ± 0.25 | - | - | - | 0.021 ± 0.005 | 1.24 ± 0.38 | 0.42 ± 0.10 |
| [22] (f.w.) | 110 ± 40 | 240 ± 30 | 900 ± 100 | 3200 ± 300 | 6.8 ± 0.5 | 2.6 ± 0.4 | 0.17 ± 0.05 | 0.02 ± 0.0 | 0.32 ± 0.11 | - | 0.007 ± 0.005 | 0.28 ± 0.08 | 0.05 ± 0.01 |
| [23] (d.w.) | - | - | - | - | 1.2 ± 0.1 | 0.8 ± 0.1 | 0.1 ± 0.0 | 0.1 ± 0.0 | <0.1 ± 0.0 | - | < 0.1 ± 0.0 | 0.3 ± 0.1 | < 0.1 ± 0.0 |
| [39] (d.w.) | - | - | - | - | 24.83 ± 1.71 | 21.72 ± 0.83 | 2.13 ± 0.21 | 3.22 ± 0.47 | 3.26 ± 0.32 | - | 0.022 ± 0.007 | - | 0.73 ± 0.21 |
| [38] (d.w.) | - | - | - | - | - | - | - | - | - | 1.56 ± 0.02 | < 0.02 | - | < 0.05 |
| [37] (f.w.) | - | - | - | - | 32.93 | 39.84 | 5.05 | 4.504 | 24.22 | - | 0.053 | - | 0.525 |
| [35] (d.w.) | - | - | - | - | 45.2 ± 2.7 | 36.2 ± 2.4 | 0.75 ± 0.05 | 3.60 ± 0.21 | 3.67 ± 0.22 | 0.15 ± 0.01 | 0.1 ± 0.01 | 1.2 | 0.28 ± 0.02 |

The turbot samples had different origins: wild turbot from the Black Sea Turkish coastline [35,37–40]; wild turbot from the Atlantic Ocean [23,41]; farmed turbot from Portugal [22].
"< LOD" = concentrations were below the limits of detection.

The Cd is found to be concentrated mainly in the intestine and liver tissues. In general, wild fish present higher Cd concentrations than farmed specimens [22]. The concentrations of Cd recorded in our present study are three times higher than the values reported by Lourenco et al. [22], a fact that confirms the aforementioned.

Ergönül and Altindağ [40] explained that the Black Sea turbot is the riskiest fish species in terms of Cr levels, but our results showed that this element did not accumulate in the turbot body, excepting the stomach tissue. Similar results were found by Plavan et al. [44], who analyzed ten different fish species from the Black Sea, and concluded that Cr had no specific tissue for accumulation.

## 4. Conclusions

The bioconcentration of essential metals in specific organs was related to their biological function. As a conclusion, it can be stated that the highest bioaccumulation capacity in terms of Ca, Mg, Na, Ni, As, Zn and Cd was registered in the caudal fin, liver and intestinal tissues. Also, other elements such as K, Fe, Cu and Mn had the highest bioaccumulation in the muscle, spleen, liver and gills tissues. The turbot male specimens accumulated a higher concentration of metals compared to females. However, this can be also influenced in a small percentage by the higher body size of males, compared to females, since they have approximately the same age. The concentrations of toxic metals in Black Sea turbot from this study were lower in the muscle samples compared to the studies conducted in Turkey, suggesting that the anthropogenic activity in the studied area did not pose a major impact on the habitat contamination. These results can be applied in the food development of turbot from aquaculture for the essential and nonessential elements' balance. Also, the results reflect the status of the Black Sea Coast and the Constanta City local environmental condition, since they do not migrate on long, transboundary distances.

**Author Contributions:** Study design and data interpretation, I.-A.S., S.-M.P. and S.-A.S.; Methodology and formal analysis, I.-A.S. and S.-A.S.; Resources V.C., M.N. and A.M.; Sampling map E.S.B. Supervision, C.F.; Writing—original draft I.-A.S., S.-M.P., S.-A.S. and C.F.

**Funding:** This research received no external funding.

**Acknowledgments:** This study was financial supported by the EU project "Doctoral students and postdoctoral researchers prepared for the labor market!" contract POCU/380/6/13/123623. The grant of the Romanian Ministry of Research and Innovation, CCCDI-UEFISCDI, project number 26PCCDI/01.03.2018,"Integrated and sustainable processes for environmental clean-up, wastewater reuse and waste valorization"(SUSTENVPRO), within PNCDI was also acknowledged. The project "EXPERT", financed by the Romanian Ministry of Research and Innovation, Contract no. 14PFE/17.10.2018 was also acknowledged.

**Conflicts of Interest:** The authors declare no conflict of interest.

**Ethical Approval:** This article does not contain any studies with human participants or laboratory animals performed by any of the authors. The fish was legally purchased.

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
