# Peer review of "Bioconcentration of Essential and Nonessential Elements in Black Sea Turbot (Psetta Maxima Maeotica Linnaeus, 1758) in Relation to Fish Gender"

_jmse, doi:10.3390/jmse7120466_

Round 1

Reviewer 1 Report

General comments:

This is an interesting study that can contribute to the field, especially as a provider of basal information. I think the article should go for a language revision to facilitate the message passing onto the readers correctly.

The introduction should be extended to better set the scene and to demonstrate the need for the study.

There is no way of knowing where did the fish came from, but the authors, to justify some results, use the reasoning that different feeding areas may have different toxicants and essential elements concentration. This, not knowing where each individual came from, is a major caveat to the work and to the understanding of the results. The authors should acknowledge this caveat and heavily discuss its implications for the study – This should be done in the discussion of the article (maybe after line 197). The authors do allude to this further into the discussion, but I think a better effort can be achieved.

Specific comments:

Line 33 - -The species name should be in italics

Lines 58-67 – Please consider moving this portion of the text to the “Methods” section. The introduction should be more general, this is specific to this work, but the reasoning for the question should be made more general.

Figure (study area map) is not numbered, I suspect it is “1”. The caption is not self-explanatory. The fishing area is delimited in grey? Please add this information to the figure caption.

Line 95 – How were the specimens identified? In the field, or in lab? What was the support literature used? Was the identification all done by the same person?

Line 98 – How were females and males identified? Only 40 individuals were bought?

Table 1 and throughout the article – I would change the “weight” to “Body mass” as the units for weight are Newtons not grams.

Lines 105-106 – Stomach content – Were all individuals’ stomach full? How did the authors progress with empty stomachs? Please refer in the text. This was partially answered in Line 138, but more information is needed here to fully understand the methods applied.

Line 107 – “wasted” or washed?

Line 150 – Please show these results on the article. Here Table 4 should be referred to (maybe the authors will need to re-numerate the tables)

Table 3, Figure 2, Table 4 – Appear before being referenced in the text.

Figure 2 – I do not fully understand this figure. What does it depict? How was it done (method and software)?

Author Response

Distinguished Editorial Border of Journal of Marine Science and Engineering,

            We would like to thank for the effort of three reviewers that helped us to increase the quality of our manuscript  with the ID jmse-648318. The corrections in the  main manuscript were done in the Word document using Track Changes option.

Reviewer # 1 specific comments: 

Line 33 - -The species name should be in italics

Answer: The correction was done.

Lines 58-67 – Please consider moving this portion of the text to the “Methods” section. The introduction should be more general, this is specific to this work, but the reasoning for the question should be made more general.

Answer: This information was moved to “Methods” section.

Figure (study area map) is not numbered, I suspect it is “1”. The caption is not self-explanatory. The fishing area is delimited in grey? Please add this information to the figure caption.

Answer: The mistake was corrected and added the explanation in the figure caption.

Line 95 – How were the specimens identified? In the field, or in lab? What was the support literature used? Was the identification all done by the same person?

Answer: The specimens were easy to be identified in Romanian territorial fishing area because there are only two commercialised species: Solea solea (sand sole) and Black Sea turbot. These are easy to be distinguished in situ having different morphological characteristics. The identification was done by same person responsible for fish purchasing.

Line 98 – How were females and males identified? Only 40 individuals were bought?

Answer: We gave more explanations in manuscript. “The fishermen allowed us to do a small abdominal incision for each fish to determine in situ the sex gender in order to reach the sampling number.”

Table 1 and throughout the article – I would change the “weight” to “Body mass” as the units for weight are Newtons not grams.

Answer: This correction was done. Thank you!

Lines 105-106 – Stomach content – Were all individuals’ stomach full? How did the authors progress with empty stomachs? Please refer in the text. This was partially answered in Line 138, but more information is needed here to fully understand the methods applied.

Answer: It was explained in the manuscript. We selected the fish with stomach content. “The fishermen allowed us to do a small abdominal incision for each fish to determine in situ the sex gender in order to reach the sampling number. This was useful to sort the fish that had stomach content.”

Line 107 – “wasted” or washed?

Answer: This correction was done. Thank you!

Line 150 – Please show these results on the article. Here Table 4 should be referred to (maybe the authors will need to re-numerate the tables)

Answer: This correction was done. Thank you! The tables were changed .

Table 3, Figure 2, Table 4 – Appear before being referenced in the text.

Answer: This correction was done. Thank you! More explanations were added.

Figure 2 – I do not fully understand this figure. What does it depict? How was it done (method and software)?

Answer: The figure illustrates the dispersion of the studied elements in the turbot body. It showed a pattern of elements localization in turbot males/females.  For data representation it was used OriginPro software.

Distinguished Editorial Border of Journal of Marine Science and Engineering,

            We would like to thank for the effort of three reviewers that helped us to increase the quality of our manuscript  with the ID jmse-648318. The corrections in the  main manuscript were done in the Word document using Track Changes option.

Reviewer # 1 specific comments: 

Line 33 - -The species name should be in italics

Answer: The correction was done.

Lines 58-67 – Please consider moving this portion of the text to the “Methods” section. The introduction should be more general, this is specific to this work, but the reasoning for the question should be made more general.

Answer: This information was moved to “Methods” section.

Figure (study area map) is not numbered, I suspect it is “1”. The caption is not self-explanatory. The fishing area is delimited in grey? Please add this information to the figure caption.

Answer: The mistake was corrected and added the explanation in the figure caption.

Line 95 – How were the specimens identified? In the field, or in lab? What was the support literature used? Was the identification all done by the same person?

Answer: The specimens were easy to be identified in Romanian territorial fishing area because there are only two commercialised species: Solea solea (sand sole) and Black Sea turbot. These are easy to be distinguished in situ having different morphological characteristics. The identification was done by same person responsible for fish purchasing.

Line 98 – How were females and males identified? Only 40 individuals were bought?

Answer: We gave more explanations in manuscript. “The fishermen allowed us to do a small abdominal incision for each fish to determine in situ the sex gender in order to reach the sampling number.”

Table 1 and throughout the article – I would change the “weight” to “Body mass” as the units for weight are Newtons not grams.

Answer: This correction was done. Thank you!

Lines 105-106 – Stomach content – Were all individuals’ stomach full? How did the authors progress with empty stomachs? Please refer in the text. This was partially answered in Line 138, but more information is needed here to fully understand the methods applied.

Answer: It was explained in the manuscript. We selected the fish with stomach content. “The fishermen allowed us to do a small abdominal incision for each fish to determine in situ the sex gender in order to reach the sampling number. This was useful to sort the fish that had stomach content.”

Line 107 – “wasted” or washed?

Answer: This correction was done. Thank you!

Line 150 – Please show these results on the article. Here Table 4 should be referred to (maybe the authors will need to re-numerate the tables)

Answer: This correction was done. Thank you! The tables were changed .

Table 3, Figure 2, Table 4 – Appear before being referenced in the text.

Answer: This correction was done. Thank you! More explanations were added.

Figure 2 – I do not fully understand this figure. What does it depict? How was it done (method and software)?

Answer: The figure illustrates the dispersion of the studied elements in the turbot body. It showed a pattern of elements localization in turbot males/females.  For data representation it was used OriginPro software.

Reviewer 2 Report

GENERAL COMMENTS

As its title suggest, the authors investigated the influence of sex gender in bioconcentration and physiological demand of essential and nonessential elements in Black Sea turbot. I found the study difficult to follow with no clear defined goals and hypotheses and most of all too long and descriptive. For example Discussion is a long piece of text comparing the concentrations found on the present study with those found on other studies, without providing any new “take home” novel messages and well-defined implications for research. There is also a lot of speculation (see specific comments below) in the text and many parts, statements from the author lack scientific support from other studies. I think the scope of the manuscript is local and lacks interest from a broader readership. Some of the table are clearly redundant and should be removed. English is very poor and the manuscript should clearly be revised by a native English speaker. Therefore, the manuscript is far from reaching an acceptable level and presently, I can only recommend rejection. I am sorry I do not have better news for these authors.

SPECIFIC COMMENTS

Line 21 – Make a brief state of the art of your manuscript, outlining current research and why your study is important.

Line 23 – What area? River, catchment, country?

Line 23-24 – “They were measured”. Who was measured??

Line 26 – How many adults? How many per group?

Line 27 – “and they were prepared a high number of samples (1200)”. English language needs to be reviewed by a native speaker: the fish were prepared a high number of samples? This 1200 refers to the name of total samples considering the two groups?

Line 28 – Do not use acronyms for these techniques upon first mention.

Line 28-29 – This seems quite simplistic. You should previously refer what was the purpose of your analysis, of your study.

Line 30-33 – Why were only 2 elements (here the full name, but previously only the acronym: please homogenize) selected out those 13? Again, authors need to clearly point the context and goals of their research.

Line 33 – How does this related to your goal outlined on line 21? i.e. what is the influence of sex gender in bioconcentration of essential and nonessential elements in turbot? This needs to be clearer.

Line 40 – Could you specify these measures?

Line 65 – Remove :

Line 66 – What do you mean by food science preservation?

Line 68 – Could you specify these elements within each group and outline their importance?

Line 69 – Why are incomplete and above all, why they are important both from the theoretical and practical point of view?

Line 70-72 – Again, you should previously point out the role of essential and non-essential elements.

Line 86 – Which biological material? The turbot? how many? weight?

Line 90 – It is therefore supposed that the biological material (fish?) are contaminated by such human pressures.

Line 91 – I think this line can be removed (don´t need to outline software to produce the maps)

Line 95 – This line can be removed.

Line 96-97 –Where were they captured and how? Were all form the same site?  What is their size-range?

Line 99 – “measured and weighted”. To the nearest cm and g?

Line 99-101 – This is a result and should not be place in this section (Material and methods).

Line 100- “for all biometric variables”. This is not true for head length (cm).

Line 101 – test instead of Test.

Line 104- Give complete captions: indicate study area, goal,..

Line 105 – What was the quantity of tissue removed in each case?

Line 116 – 30 samples x 40 fish? If so, please include.

Line 118-119 – equipment brand and model should be in parenthesis.

Line 122 – Why do you need to validate the methods?

Line 123 – “ERM-BB422”. What does this mean?

Line 124 – Are these other fish other than the 40? replace “replicas” by “replicates”.

Line 125 – Again, Results should not be presented in the Material and Methods section.

Table 2 – What do you mean by confidence? Is it the standard deviation (SD) as you have on Table 1? Be consistent!

Line 129 – Did you also test your data for homoscedasticity? What kind of data was tested?

Line 134 – Was it not the same as you did with the ANOVA?

Line 137 – Could you provide a reference and a brief explanation for this index?

Line 139 – Any evidence to support this?

Line 140-141 – This is not clear and needs to be better explained. Do you have a reference/study to support this index?

Line 144 – LOD?

Table 3 – Use the same number of decimals for each pair of entries (i.e. males and females).

Line 161-162 – It seems there is some speculation here; there is no references/previous studies to support this.

Fig. 2 (lines 175-179) – What’s the name of this analysis? I do not see it adds much to the results on Table 3.

Line 179-180 – This sentence is unclear. The results of the ANOVA are the test statistic and significance value and not “physiological demand and bioconcentration affinity the turbot body.”

Table 4 – This table is redundant to Table 3, as it a simple ordination (from higher to lower) of the concentrations in each organ. What does the table show indeed’ It says only “The results of multiple comparisons…”

Line 186 – “In this case the stomach content was used to measure it.”. Could you clarify?

Line 187 – “The analysis of the bioaccumulation from sediments was considered to be not entirely correct”. This is a concern if you are doubting of the quality of your analyses.

Line 190 – This should be said earlier. Same on line 193

Line 191 – Again this is pure speculation and should be removed. Same on line 222

Line 194 – What evidence from the Table show it was different? Did you make in statistical test?

Line 195-196 – I do not understand what the authors mean with this, but this was not said previously In Material and methods.

Line 200 – What papers? Specify.

Line 203 – “Most of the studies were concerned on this subject because turbot meat has economic value”. You said previously that there were few studies (see abstract and Intro).

Line 241 – Both. Which ones? You only mention one, Manthey-Karl et al. [55].

Line 250 – This is too vague and requires further explanations/support.

Line 259 – The units in your results are different from the ones of Lourenço et al.

Line 293- Previously you call essential and non-essential elements, now you call them macro- and microelements. Be consistent.

Table 6 – This analysis was not referred on Material and methods as well as its purpose. Please include it.

Line 298 – “were very fast concentrated”. How do you know it was fast?

Line 300 – Where is the map shown?

Line 301-302 – again there is speculation here that should be removed.

Line 303-304 – Any reference to support this?

Line 304-306-  This seems logical and hence, the authors are now showing any novelty here.

Line 306 – Again, these isolation and speculative sentences with no support, should be removed.

Author Response

Reviewer # 2 specific comments:

Line 21 – Make a brief state of the art of your manuscript, outlining current research and why your study is important.

Line 23 – What area? River, catchment, country? Line 23-24 – “They were measured”. Who was measured?? Line 26 – How many adults? How many per group?

Line 27 – “and they were prepared a high number of samples (1200)”. English language needs to be reviewed by a native speaker: the fish were prepared a high number of samples? This 1200 refers to the name of total samples considering the two groups?

Line 28 – Do not use acronyms for these techniques upon first mention. Line 28-29 – This seems quite simplistic. You should previously refer what was the purpose of your analysis, of your study.

Line 30-33 – Why were only 2 elements (here the full name, but previously only the acronym: please homogenize) selected out those 13? Again, authors need to clearly point the context and goals of their research.

Line 33 – How does this related to your goal outlined on line 21? i.e. what is the influence of sex gender in bioconcentration of essential and nonessential elements in turbot? This needs to be clearer.

Line 40 – Could you specify these measures? Line 65 – Remove : Line 66 – What do you mean by food science preservation?

Line 68 – Could you specify these elements within each group and outline their importance?

Line 69 – Why are incomplete and above all, why they are important both from the theoretical and practical point of view?

Line 70-72 – Again, you should previously point out the role of essential and non-essential elements.

Line 86 – Which biological material? The turbot? how many? weight? Line 90 – It is therefore supposed that the biological material (fish?) are contaminated by such human pressures.

Line 91 – I think this line can be removed (don´t need to outline software to produce the maps) Line 95 – This line can be removed.

Line 96-97 –Where were they captured and how? Were all form the same site? What is their size-range?

Line 99 – “measured and weighted”. To the nearest cm and g? Line 99-101 – This is a result and should not be place in this section (Material and methods). Line 100- “for all biometric variables”. This is not true for head length (cm). Line 101 – test instead of Test.

Line 104- Give complete captions: indicate study area, goal,..

Line 105 – What was the quantity of tissue removed in each case?

Line 116 – 30 samples x 40 fish? If so, please include.

Line 118-119 – equipment brand and model should be in parenthesis. Line 122 – Why do you need to validate the methods?

Line 123 – “ERM-BB422”. What does this mean? Line 124 – Are these other fish other than the 40? replace “replicas” by “replicates”.

Line 125 – Again, Results should not be presented in the Material and Methods section. Table 2 – What do you mean by confidence? Is it the standard deviation (SD) as you have on Table 1? Be consistent! Line 129 – Did you also test your data for homoscedasticity? What kind of data was tested? Line 134 – Was it not the same as you did with the ANOVA?

Line 137 – Could you provide a reference and a brief explanation for this index? Line 139 – Any evidence to support this?

Line 140-141 – This is not clear and needs to be better explained. Do you have a reference/study to support this index?

Line 144 – LOD? Table 3 – Use the same number of decimals for each pair of entries (i.e. males and females).

Line 161-162 – It seems there is some speculation here; there is no references/previous studies to support this. 2 (lines 175-179) – What’s the name of this analysis? I do not see it adds much to the results on Table 3.

Line 179-180 – This sentence is unclear. The results of the ANOVA are the test statistic and significance value and not “physiological demand and bioconcentration affinity the turbot body.” Table 4 – This table is redundant to Table 3, as it a simple ordination (from higher to lower) of the concentrations in each organ. What does the table show indeed’ It says only “The results of multiple comparisons…”

Line 186 – “In this case the stomach content was used to measure it.”. Could you clarify? Line 187 – “The analysis of the bioaccumulation from sediments was considered to be not entirely correct”. This is a concern if you are doubting of the quality of your analyses.

Line 190 – This should be said earlier.

Same on line 193 Line 191 – Again this is pure speculation and should be removed.

Same on line 222 Line 194 – What evidence from the Table show it was different? Did you make in statistical test? Line 195-196 – I do not understand what the authors mean with this, but this was not said previously In Material and methods.

Line 200 – What papers? Specify. Line 203 – “Most of the studies were concerned on this subject because turbot meat has economic value”. You said previously that there were few studies (see abstract and Intro).

Line 241 – Both. Which ones? You only mention one, Manthey-Karl et al. [55]. Line 250 – This is too vague and requires further explanations/support.

Line 259 – The units in your results are different from the ones of Lourenço et al. Line 293- Previously you call essential and non-essential elements, now you call them macro- and microelements. Be consistent. Table 6 – This analysis was not referred on Material and methods as well as its purpose. Please include it.

Line 298 – “were very fast concentrated”. How do you know it was fast?

Line 300 – Where is the map shown? Line 301-302 – again there is speculation here that should be removed.

Line 303-304 – Any reference to support this? Line 304-306- This seems logical and hence, the authors are now showing any novelty here. Line 306 – Again, these isolation and speculative sentences with no support, should be removed.

Answer: The corrections were done according to these suggestions.

Reviewer 3 Report

It is important to add information about the age of specimens, their maturity and their gonad consistency.

Also, it is important to know details concerning migration of Black Sea turbot, especially in spring time when the samples were obtained. Are this fish sedentary and really reflect local environment condition or are migrative and reflect toxic condition in some other area of the Black Sea.

Compared male and female specimens significantly differ by their body size. This is very important, and authors certainly should discuss obtained results from this point of view – are the difference in bioconcentrations really sex specific or they are age/size specific.

I also suggest to change the title of paper and delete “… physiological demand …” because physiological demand of metals was not studied.

Author Response

Reviewer # 3 specific comments:

It is important to add information about the age of specimens, their maturity and their gonad consistency. Also, it is important to know details concerning migration of Black Sea turbot, especially in spring time when the samples were obtained. Are this fish sedentary and really reflect local environment condition or are migrative and reflect toxic condition in some other area of the Black Sea. Compared male and female specimens significantly differ by their body size. This is very important, and authors certainly should discuss obtained results from this point of view – are the difference in bioconcentrations really sex specific or they are age/size specific. I also suggest to change the title of paper and delete “… physiological demand …” because physiological demand of metals was not studied.

Answer: All corrections were done according to these suggestions. Thank you!

Reviewer # 3 specific comments:

It is important to add information about the age of specimens, their maturity and their gonad consistency. Also, it is important to know details concerning migration of Black Sea turbot, especially in spring time when the samples were obtained. Are this fish sedentary and really reflect local environment condition or are migrative and reflect toxic condition in some other area of the Black Sea. Compared male and female specimens significantly differ by their body size. This is very important, and authors certainly should discuss obtained results from this point of view – are the difference in bioconcentrations really sex specific or they are age/size specific. I also suggest to change the title of paper and delete “… physiological demand …” because physiological demand of metals was not studied.

Answer: All corrections were done according to these suggestions. Thank you!

Round 2

Reviewer 2 Report

Though there seems to be some improvements over the original submission, I could not even assess how all my concerns were dealt with.

Indeed, it was quite disappointing to see that, in response to all of my concerns, the authors simply replied "The corrections were done according to these suggestions.", instead of replying on a point-by-point basis as it is required by the author guidelines, which I quote: "all reviewer comments should be responded to in a point-by-point fashion. Where the authors disagree with a reviewer, they must provide a clear response".

So, please provide your answers on a point-by point basis to each of the comments.

Further, English language is still poor and lacks revision by a native English speaker.

Author Response

Distinguished Editorial Border of Journal of Marine Science and Engineering,

To Reviewer # 2,

            First of all, I would like to thank you for all your effort in reviewing this paper and also, to present apologies for our previous not consistent response for your 1st report.   

            Reviewer # 2 specific comments:

Line 21 – Make a brief state of the art of your manuscript, outlining current research and why your study is important.

Answer: I have integrated into the previous introduction the following paragraphs:

As the Black Sea is considered a semi-enclosed sea and has positive freshwater balance, mainly due to high river inputs, it is exposed to heavy metals contamination, especially in coastal area. Fish are considered top consumers in all aquatic systems and therefore, they are continuously exposed to contaminants, situation which involves tissues bioaccumulation especially in case of demersal species, of both essential and non-essential heavy metals. Thus, as elements have the tendency to accumulate in specific hot spots, it is recommended the use of demersal fish species as bio-indicators, in order to evaluate the heavy metals pollution. Thus, analyzing different fish tissues ensures scanning of aquatic environment, considering fish mobility and capacity to continuously accumulate xenobiotics.

A series of demersal fish species as turbot (Psetta maxima maeotica) and whiting (Merlangius merlangus) can be used as bioindicators in order to evaluate the heavy metals pollution in certain coastal areas, since they are species that do not migrate on long, transboundary distances [25]. Also, there species are involved only in local migrations within the coastal areas for spawning, feeding and wintering [25].

Therefore, the analysis of demersal fish catches in terms of heavy metals concentrations can represent an important instrument in order to evaluate the coastal areas pollution, especially after wintering period, when fish stock migrate closer to shore, at water depths between 20-40m, for spawning.

Among Black Sea demersal fish species, turbot presents interest in research for the biomonitoring of polychlorinated biphenyls [26, 27]; food science preservation – freshness assessment of turbot by using different biochemical and proteomics methods [28]; molecular genetics [29]; ecology and behaviour [30]; biomarker in oil pollution [31]; nitrite toxicity [32, 33]; aquaculture [34]; heavy metal biomonitoring [35-40]. Several research papers conducted by Ergönül and Altindağ, 2014 [40]; Nisbet et al., 2010 [39]; Das et al., 2009 [38]; Bat et al., 2006 [37]; Tuzen 2003, 2009 [26, 36] studied the concentration of potentially toxic metals and metalloids (Zn, Fe, Cu, Ni, Mn, Cr, Cd, Pb, As) in the Black Sea turbot muscle tissue captured in Turkish territorial marine waters. Other scientific articles realised by Manthey-Karl et al., 2016 [41], Martinez et al., 2010 [23] and Lourenco et al., 2012 [22] studied the concentration of essential macro-elements (Ca, Mg, Na, K), micro-elements (Zn, Fe, Cu, Ni, Mn) and potentially toxic metals (Cd, Pb, As) in the muscle tissue of wild turbot captured in Atlantic Ocean, respectively, in farmed turbot from Portugal. However, the studies focused on essential, non-essential elements and toxic metals in the Black Sea turbot organism are incomplete since only potential toxic micro-elements and trace-elements were determined The macro-elements are important to be determined in order to identify their capacity of influence micro and trace-elements bioaccumulation. Also, those studies can contribute to a better evaluation of certain Black Sea coastal areas, considered under a continuous anthropogenic pressure.

This study investigates the influence of turbot gender, catched in the Romanian Black Sea Coast, in bioconcentration of both essential (Ca, Cu, Fe, K, Mg, Mn, Na, Zn) and nonessential elements (As, Cd, Ni) in various tissues (gills - Gi, stomach - St, intestine - In, liver - Lv, spleen - Sp, gonads - Go, muscles - Mu, caudal fin - Cf), including stomach and intestine content. This evaluation will also contribute to an upgraded characterization of Romanian Black Sea Coast, in terms of heavy metals pollution.

Line 23 – What area? River, catchment, country?

Answer: This study investigates the influence of gender in bioconcentration of essential and nonessential elements in different parts of Black Sea turbot (Psetta maxima maeotica) body, from an area considered under high anthropogenic pressure (the Constanta city Black Sea Coastal Area - Romania).

Line 23-24 – “They were measured”. Who was measured??

Answer: A number of 13 elements (Ca, Mg, Na, K, Fe, Zn, Mn, Cu, Ni, Cr, As, Pb and Cd) were measured in various sample types: muscle, stomach, stomach content, intestine, intestine content, gonads, liver, spleen, gills and caudal fin.

Line 26 – How many adults? How many per group? Line 27 – “and they were prepared a high number of samples (1200)”. English language needs to be reviewed by a native speaker: the fish were prepared a high number of samples? This 1200 refers to the name of total samples considering the two groups? Line 28 – Do not use acronyms for these techniques upon first mention.

Answer for aforementioned comments: Turbot adults (4-5 years old) were separated, according to their gender, in two groups (20 males, respectively 20 females) and a high total number of samples (1200 from both groups) were prepared and analysed, in triplicate, with Flame Atomic Absorption Spectrometry and high-resolution continuum source atomic absorption spectrometry with graphite furnace techniques.

Line 28-29 – This seems quite simplistic. You should previously refer what was the purpose of your analysis, of your study.

Answer: The results were statistical analysed in order to emphasize the bioconcentration of the determined elements in different tissues of wild turbot males vs. females and also, to contribute to an upgraded characterization of Romanian Black Sea Coast – Constanta city, in terms of heavy metals pollution.

Line 30-33 – Why were only 2 elements (here the full name, but previously only the acronym: please homogenize) selected out those 13? Again, authors need to clearly point the context and goals of their research. Line 33 – How does this related to your goal outlined on line 21? i.e. what is the influence of sex gender in bioconcentration of essential and nonessential elements in turbot? This needs to be clearer.

Answer for aforementioned comments: The Mg and Zn have different roles in gonads of males and females, as they were the only elements with completely different patterns between the analyzed groups of specimens. The concentrations of studied elements in muscle were not similar with data provided by literature, suggesting that chemistry of the habitat and food plays a major role in the availability of the metals in the body of analyzed fish species. The gender influenced the bioaccumulation process of all analyzed elements in most tissues since turbot male specimens accumulated higher concentration of metals compared to females.  The highest bioaccumulation capacity in terms of Ca, Mg, Na, Ni, As, Zn and Cd was registered in caudal fin, liver and intestine tissues. Also, other elements such as K, Fe, Cu and Mn had the highest bioaccumulation in muscle, spleen, liver and gills tissues.

Line 40 – Could you specify these measures?

Answer: On the other hand, nonessential metals, such as toxic metals resulted in the water from fine suspended solids in the water, may alter the feeding rate of fish and determine a reduction in the metabolic efficiency [3, 4]. Sublethal effects manifested when fish are exposed to acute metals concentrations include damage of sensory organs and receptors, which leads to impairment of sensorial perception (namely olfaction) [4] According to Gati et al., 2016, sediment samples with a higher value than 4 of the probable effect concentration quotient (PEC-Q) are more likely to have acute or toxic effects on benthic organisms [3]. In this way, integrated measures for toxicological risk ranking in fish communities were developed [5].

Line 65 – Remove: Line 66 – What do you mean by food science preservation?

Answer for aforementioned comments:  Among Black Sea demersal fish species, turbot presents interest in research for the biomonitoring of polychlorinated biphenyls [26, 27]; food science preservation – freshness assessment of turbot by using different biochemical and proteomics methods [28]; molecular genetics [29]; ecology and behaviour [30]; biomarker in oil pollution [31]; nitrite toxicity [32, 33]; aquaculture [34]; heavy metal biomonitoring [35-40].

Line 68 – Could you specify these elements within each group and outline their importance? Line 69 – Why are incomplete and above all, why they are important both from the theoretical and practical point of view?

Answer for aforementioned comments: Several research papers conducted by Ergönül and Altindağ, 2014 [40]; Nisbet et al., 2010 [39]; Das et al., 2009 [38]; Bat et al., 2006 [37]; Tuzen 2003, 2009 [26, 36] studied the concentration of potentially toxic metals and metalloids (Zn, Fe, Cu, Ni, Mn, Cr, Cd, Pb, As) in the Black Sea turbot muscle tissue captured in Turkish territorial marine waters. Other scientific articles realised by Manthey-Karl et al., 2016 [41], Martinez et al., 2010 [23] and Lourenco et al., 2012 [22] studied the concentration of essential macro-elements (Ca, Mg, Na, K), micro-elements (Zn, Fe, Cu, Ni, Mn) and potentially toxic metals (Cd, Pb, As) in the muscle tissue of wild turbot captured in Atlantic Ocean, respectively, in farmed turbot from Portugal. However, the aforementioned studies focused on essential, non-essential elements and toxic metals in the Black Sea turbot organism are incomplete since only potential toxic micro-elements and trace-elements were determined. The macro-elements are important to be determined in order to identify their capacity of influence micro and trace-elements bioaccumulation. Also, those studies can contribute to a better evaluation of certain Black Sea coastal areas, considered under a continuous anthropogenic pressure.

Line 70-72 – Again, you should previously point out the role of essential and non-essential elements.

Answer: This study investigates the influence of turbot gender, catched in the Romanian Black Sea Coast, in bioconcentration of both essential (Ca, Cu, Fe, K, Mg, Mn, Na, Zn) and nonessential elements (As, Cd, Ni) in various tissues (gills - Gi, stomach - St, intestine - In, liver - Lv, spleen - Sp, gonads - Go, muscles - Mu, caudal fin - Cf), including stomach and intestine content. This evaluation will also contribute to an upgraded characterization of Romanian Black Sea Coast, in terms of heavy metals pollution.

Line 86 – Which biological material? The turbot? how many? weight? Line 90 – It is therefore supposed that the biological material (fish?) are contaminated by such human pressures. Line 91 – I think this line can be removed (don´t need to outline software to produce the maps) Line 95 – This line can be removed. Line 96-97 –Where were they captured and how? Were all form the same site? What is their size-range? Line 99 – “measured and weighted”. To the nearest cm and g? Line 99-101 – This is a result and should not be place in this section (Material and methods). Line 100- “for all biometric variables”. This is not true for head length (cm). Line 101 – test instead of Test. Line 104- Give complete captions: indicate study area, goal,..

Answer for aforementioned comments:  The area of study was chosen since it is the most intensely populated from the Romanian coastline and also, the main anthropogenic activities (tourism, municipal wastewater effluents, industry, cargo shipping and oil refinery) are concentrated here [43, 46]. The map of the Romanian Black Sea targeted coastal area is presented in figure 1.

Since fish stocks are affected by high anthropogenic pressure, the biological material in present study is represented by a number of 40 Black Sea turbot specimens. The biological material was selected from a larger number of turbot specimens purchased fresh from the fish market located in Constanta City and surrounding area, in April 2016 (Fig. 1). All turbot specimens were catched in the fishing areas along the coastline of Constanta City, by using bottom (turbot) gill nets with a minimum mesh size of 180 mm. Flatfish are considered to be suitable bioindicators of pollution due to their ecological importance, sensitivity and close contact to sediment substrates [47].

The studied material was taxonomically identified as Psetta maxima maeotica Linnaeus, 1758, by using both an online fish taxonomically identification database [51] and a taxonomic identifier for Black Sea fish species [52]. The gender identification was made in situ since we had the permission to do a small abdominal incision, with plastic laboratory instruments, for each fish, to determine the sex gender, in order assure the targeted sampling number for this study (20 males and 20 females). This was also useful for the identification and selection of the fish specimens that had stomach content. A number of 40 turbot specimens (20 males and 20 females) were placed in polyethylene bags, stored on ice and transported to the laboratory, where they were frozen at -20°C, until the samples were prepared and analysed. Before the analysis, the specimens were dissected by using plastic laboratory instruments, for avoiding samples contamination. Each individual was biometric measured and individual biomass was determined before frozen at -20°C since, according to different authors, the fish size can influence the bioaccumulation process [53-55]. The results are presented in Table 1. In this study, the males group had significant (*p<0.05) higher values than the female group for all biometric variables according to t-test results. – it was removed in the results and discussions section, together with table 1.  Based on biometric measurements, all specimens were considered as part of a cohort of 4-5 years old adults, capable for reproduction, according to Maximov et al. [56], that collected samples, in its scientific study, from similar area – Black Sea Coast - Constanta City. The age of turbot specimens was determined according to Eryilmaz and Dalyan method [57]

Table 1. Biometric measurements and individual biomass of the studied fish (mean±SD).  – head length average values were reversed because they were wrong placed previously

Gender

Biometric measurement

Female

Male

Total length (cm)

44.60±1.86

48.3±1.55

Maximum width (cm)

32.6±2.15

35.1±0.28

Individual biomass (kg)

1.83±0.29

2.32±0.12

Distance between eyes (cm)

1.4±0.05

1.5±0.02

Caudal fin length (cm)

8.24±0.05

10.18±0.3

Head length (cm)

8.74±0.67

10.21±0.49

Head maximum width (cm)

8.9±0.02

12.56±0.56

Line 105 – What was the quantity of tissue removed in each case?

Answer: After dissection, the following sample types were collected (1 g each) for analysis: muscle, stomach, stomach content, intestine, intestine content, gonads, liver, spleen, gills and caudal fin. All the analysed specimens had over 70% stomach content, reported to the entire stomach capacity.

Line 116 – 30 samples x 40 fish? If so, please include.

Answer: The total amount of prepared and analysed samples in this study was equal to 1200 (10 samples per each fish specimen, in triplicate).

Line 118-119 – equipment brand and model should be in parenthesis.

Answer: For the metal quantification analyses two methods were used: flame atomic absorption spectrometer FL-AAS(GBC Avanta, Australia), for calcium (Ca), magnesium (Mg), sodium (Na), potassium (K), iron (Fe), zinc (Zn) and high resolution continuum source atomic absorption spectrometer with graphite furnace HR-CS GF-AAS (equipment ContrAA 600-Analytik Jena, Germany), for copper (Cu), manganese (Mn), nickel (Ni), arsenic (As), cadmium (Cd), lead (Pb), chromium (Cr).

Line 122 – Why do you need to validate the methods? Line 123 – “ERM-BB422”. What does this mean?

Answer:  Certified reference materials (CRMs) are recognized to be an essential tool in assuring the accuracy and establishing the traceability of the results of measurements [59]. Therefore, the reference material for fish muscle (ERM- BB422 type certified reference material) was analysed, certified by the Joint Research Center Institute for Reference Materials and Measurements.

Line 124 – Are these other fish other than the 40? replace “replicas” by “replicates”. Line 125 – Again, Results should not be presented in the Material and Methods section. Table 2 – What do you mean by confidence? Is it the standard deviation (SD) as you have on Table 1? Be consistent!

Answer for aforementioned comments:  The aforementioned reference material was prepared in 6 replicates, following the same protocol as a normal sample [44], in order to eliminate any determination errors and the results obtained were presented in Table 2. The certificate of analysis contained the following elements: As, Cd, Cu, Fe, Hg, I, Mn, Se, Zn, Ca, Cl, K, Mg, Na.  – in our opinion, this is part of material and methods section since it is a step that assures the accuracy and establish the traceability of the results of measurements.

Table 2. Method for assuring the accuracy and establishing the traceability of the measurements results, based on the use of CRM (µg g-1 dry weight). – I have change confidence with SD

Measured

element

Certified value mean±SD

 (µg g-1 )

Measured value mean±SD

 (µg g-1 )

Analytical method

Number of replicates

As

12.7±0.7

11.3±0.4

HR-CS GF-AAS

6

Ca

342

340±6

FL-AAS

6

Cd

0.0075±0.0018

0.0077±0.002

HR-CS GF-AAS

6

Cu

1.67±0.16

1.62±0.27

HR-CS GF-AAS

6

Fe

9.4

9.5±0.3

FL-AAS

6

K

21400

21381±21

FL-AAS

6

Mg

1370

1372±7

FL-AAS

6

Mn

0.368±0.028

0.361±0.034

HR-CS GF-AAS

6

Na

2800

2771±27

FL-AAS

6

Zn

16±1.1

15±1.7

FL-AAS

6

Line 129 – Did you also test your data for homoscedasticity? What kind of data was tested? Line 134 – Was it not the same as you did with the ANOVA?

Answer for aforementioned comments: We did not do homoscedasticity tests because we considered that ANOVA, applied for analyzing data, is based on the normality assumptions of error component (random component or residual component). Accordingly, it is not necessary to the residuals for normality and homogeneity of variance.

First, the Levene’s homogeneity variance test andthe Kolmogorov-Smirnov normality test were performed to study the data distribution in both experimental groups (males and females). The result of each test proved that the data had a normal distribution. After this step they were performed one-way ANOVA for analysis of the variance significance for each element concentration between tissues that were sampled from both genders, followed by multiple comparisons Tukey HSD test, in order to determined which of the means are different. The t-Test was performed to observe any significant differences between females and males regarding all the concentrations of studied elements for each sample type. All statistical analysis was carried out using OriginPro v.9.3. (2016) software (OriginLab Corporation, USA). All the data for metal concentration were presented in this study as average±SD. The mean values registered for different metals concentrations in the muscle tissue were centralised and compared to data reported in other studies, which sampled muscle tissue from aquaculture and wild turbot.

Line 137 – Could you provide a reference and a brief explanation for this index?

Answer: The bioconcentration factor was calculated for dietary exposure according to the formula (1) [60].

Line 139 – Any evidence to support this?

Answer: The main route of absorption of the elements in the body was considered to be the food and sediments that were ingested, according to Bury et al. who pointed out that the diet is the main source of essential metals in the fish body [61].

Line 140-141 – This is not clear and needs to be better explained. Do you have a reference/study to support this index?

Answer: The value of this factor indicates the capacity of a specific biological sample to accumulate a certain part from the total element input in the fish body. The index was used also in other similar studies [61].

Line 144 – LOD?

Answer: The Pb and Cr were below the detection limit of quantification (LOQ) for the calibration curve method (LOQ Pb-0.032 µg L-1, LOQ Cr-0.3 µg L-1) in all analyzed samples, excepting for the stomach samples where was measured Cr.

Table 3 – Use the same number of decimals for each pair of entries (i.e. males and females). Line 161-162 – It seems there is some speculation here; there is no references/previous studies to support this.

Answer: This could be related to turbot diet and its mobility in various areas with different elements concentration, where the food resources are located [25].

2 (lines 175-179) – What’s the name of this analysis? I do not see it adds much to the results on Table 3.

Answer: Figure 2. The representation of elements affinity and variation in the body of male and female specimens of Black Sea turbot, based on the registered results (µg g-1 fresh weight- f.w.). (one-way ANOVA analysis was applied for males and females in order to determine the variation of each studied element ;*p<0.05).

The figure 2 emphasizes the variations and affinity of elements in the body of male and female specimens of Black Sea turbot, based on the registered results recorded from the analysed sample tissues. It must be noted that the studied males were bigger than the females. This can be one of the explanations and the role of these elements in gonads.

Line 179-180 – This sentence is unclear. The results of the ANOVA are the test statistic and significance value and not “physiological demand and bioconcentration affinity the turbot body.” Table 4 – This table is redundant to Table 3, as it a simple ordination (from higher to lower) of the concentrations in each organ. What does the table show indeed’ It says only “The results of multiple comparisons…”

Answer: We have deleted both table 4 and the ,, The results of the ANOVA are the test statistic and significance value and not “physiological demand and bioconcentration affinity the turbot body.”

Line 186 – “In this case the stomach content was used to measure it.”. Could you clarify? Line 187 – “The analysis of the bioaccumulation from sediments was considered to be not entirely correct”. This is a concern if you are doubting of the quality of your analyses. Line 190 – This should be said earlier. Same on line 193 Line 191 – Again this is pure speculation and should be removed. Same on line 222 Line 194 – What evidence from the Table show it was different? Did you make in statistical test? Line 195-196 – I do not understand what the authors mean with this, but this was not said previously In Material and methods.

Answer for aforementioned comments:  The bioconcentration factor of the essential and nonessential elements in the Black Sea turbot was calculated from the dietary intake. In this case, the dietary intake is represented by the stomach content. It must be highlighted that if bioaccumulation analysis is conducted in relation to sediments, the results could be not entirely correct since Black Sea turbot specimens are mobile benthic organisms. The chemistry of Black Sea sediments differs from an area to another [46] and the location where the captured specimens lived cannot be identified exactly since turbot is capable to travel significant distances in search for food [25]. Thus, in the present study it has been chosen to calculate BCF of non-essential and essential elements, for all analysed tissues, in relation to stomach content samples. In table 4 were presented the values of bioconcentration factor calculated from dietary exposure, both for male and female specimens. Each element manifests a bioconcentration propensity for a specific organ. The capacity trend of bioconcentration was same for males and females, excepting the gonads where statistically different results, (p<0.05) – ANOVA, followed by Tukey HSD test, were recorded. The exposure history for each specimen and the area from which they come from are both unknown.

Line 200 – What papers? Specify. Line 203 – “Most of the studies were concerned on this subject because turbot meat has economic value”. You said previously that there were few studies (see abstract and Intro).

Answer for aforementioned comments:  The role of essential and nonessential elements on various fish species was described by scientific papers [62, 63]. Not all of the elements were able to be studied, situation that generates incomplete answers. Our results were compared with studies regarding the essential and nonessential elements from muscle samples of farmed turbot [22] or wild turbot from Atlantic Ocean [23, 41]. The scientific studies on turbot were driven by high economic value of turbot meat. Thus, not all the elements were investigated, in detail.

Line 241 – Both. Which ones? You only mention one, Manthey-Karl et al. [41].

Answer: The Na is known to be involved in important physiological activities, such as sodium channels, and in the osmotic pressure regulation [66]. The blood of a marine fish contains salt at a concentration of one-third of that of full-strength seawater, thus, fish drink a specific amount of water (0.5% body weight/hour) to regulate the osmotic pressure [67]. The mean Na values in the turbot muscles are similar with the results obtained by Manthey-Karl et al. [41], explained by the fact that both studies (present and aforementioned) analysed samples fromwild turbot specimens. The Na pathway in the body of analysed turbot specimens did not significantly differ between genders, even if males concentrated more Na in the caudal fin, stomach content and gills, compare to females. The main route of NaCl uptake is the intestine, while gills represent the main excretion route [67].

Line 250 – This is too vague and requires further explanations/support.

Answer: Zinc mean concentrations in our turbot muscles samples were similar to the results obtained by Ergönül and Altindağ [40] for wild turbot from the Black Sea. Twice higher Zn concentrations were registered in turbot muscle, compared to the results recorded by Lourenco et al [22] and Manthey-Karl et al [41]. Also, the Zn values of Bat et al. [37] and Tuzen [36] (which also analysed Black Sea turbot specimens) were two, respectively three times higher in the turbot muscles, compared to the concentrations registered in present study. This may be due to the applied analysis methodology (the aforementioned studies did not use CRM for assuring the accuracy and establishing the traceability of the measurements results) or sampling area (the Black Sea Turkish coast has been observed to be more polluted in terms of heavy metals, compared to the the Romanian coast [68].

Line 259 – The units in your results are different from the ones of Lourenço et al.

Answer: According to Lourenco et al. [22], Fe was present in concentrations of approximately 5.0 µg g-1 in marine fish species, which was similar to present study results. – Lourenço et al. used mg kg -1, that is equivalent with µg g-1. Therefore, I have changed only the units and the concentration value remains the same. 

Line 293- Previously you call essential and non-essential elements, now you call them macro- and microelements. Be consistent.

Answer: Essential and non-essential elements concentrations in the turbot muscle (mean±SD) µg g-1, f.w.-fresh weight, d.w.-dry weight.

Table 6 – This analysis was not referred on Material and methods as well as its purpose. Please include it.

Answer: Table 5 presents a comparative study between mean concentrations of elements recorded in present study and the concentrations reported, in their research, by other authors. The comparative study revealed a high variability related to essential and non-essential elements concentrations in turbot specimens from the Black Sea and other turbot subspecies, from different parts of the world. So far, this study covers the lack of published data in present area of interest, between the years 2006-2016.

Line 298 – “were very fast concentrated”. How do you know it was fast? Line 300 – Where is the map shown? Line 301-302 – again there is speculation here that should be removed. Line 303-304 – Any reference to support this? Line 304-306 - This seems logical and hence, the authors are now showing any novelty here. Line 306 – Again, these isolation and speculative sentences with no support, should be removed.

Answer for aforementioned comments:  The bioconcentration of essential metals in specific organs was related with their biological function. As a conclusion, it can be stated that the highest bioaccumulation capacity in terms of Ca, Mg, Na, Ni, As, Zn and Cd was registered in caudal fin, liver and intestine tissues. Also, other elements such as K, Fe, Cu and Mn had the highest bioaccumulation in muscle, spleen, liver and gills tissues. The turbot male specimens accumulated higher concentration of metals compared to females. However, this can be also influence in a small percentage, by the higher body size of males, compared to females, since they have approximately the same age. The concentrations of toxic metals in Black Sea turbot from this study were lower in the muscle samples compared with the studies conducted in Turkey suggesting that the anthropogenic activity in the studied area had not a major impact on the habitat contamination. These results can be applied in food development of turbot from aquaculture for essentials and non-essential elements balance. Also, the results reflect the status of Black Sea Coast – Constanta city local environmental condition, since they do not migrate on long, transboundary distances.

The underline text is meant to serve as additional information for you and those comments are not included in the present paper. The rest of the answers, that are not underlined are included in the manuscript.

Also, we have revised the English language of the present paper.

Thank you again for your understanding and we hope we have fulfilled all the specified notes, comments and suggestions from your review report.

Song, Y. F.; Luo, Z.; Huang, C.; Liu, X.; Pan, Y. X.; Chen, Q. L. Effects of calcium and copper exposure on lipogenic metabolism, metal element compositions and histology in Synechogobius hasta. Fish Physiol Biochem. 2013, 39, 1641–1656, DOI 10.1007/s10695-013-9816-4 Santos, I.; Diniz, M.S.; Carvalho, M.L.; Santos, J.P. Assessment of essential elements and heavy metals content on Mytilus galloprovincialis from River Tagus estuary. Biol. Trace. Elem. Res. 2014, 159, 233-240, doi:10.1007/s12011-014-9974-y Gati, G.; Pop, C.; Brudasca, F.; Gurzau, A.E.; Spinu, M. The ecological risk of heavy metals in sediment from the Danube Delta. Ecotoxicology. 2016, 25, 688-696, doi:10.1007/s10646-016-1627-9 Castro, B.B.; Sobral, O.; Guilhermino, L.; Ribeiro, R. An in-situ bioassay integrating individual and biochemical responses using small fish species. Ecotoxicology. 2004, 13, 667-681, doi:10.1007/s10646-003-4427-y Hartwell, S.I.; Dawson, C.E.; Durell, E.Q.; Alden, R.W.; Adolphson, P.C.; Wright, D.A.; Coelho, G.M.; Magee, J.A. Integrated measures of ambient toxicity and fish community diversity in Chesapeake Bay tributaries. Ecotoxicology. 1998, 7, 19-35 Lino, A.S.; Galvão, P.M.A.; Longo, R.T.L.; Azevedo-Silva, C.E.; Dorneles, P.R.; Torres, J.P.M.; Malm, O. Metal bioaccumulation in consumed marine bivalves in Southeast Brazilian coast. J Trace Elem in Med Bio. 2016, 34, 50-55, http://dx.doi.org/10.1016/j.jtemb.2015.12.004 Capillo, G.; Silvestro, S.; Sanfilippo, M.; Fiorino, E.; Giangrosso G.; Ferrantelli V.; Vazzana I.; Faggio C. Assessment of electrolytes and metals profile of the Faro Lake (Capo Peloro Lagoon, Sicily, Italy) and its impact on Mytilus galloprovincialis. Chemestry & Biodiversity, 2018.doi 10.1002/cbdv.201800044; Aliko, V.; Qirjo, M.; Sula, E.; Morina ,V.; Faggio, C. Antioxidant defense system, immune response and erythron profile modulation in Gold fish, Carassius auratus, after acute manganese treatment. Fish Shellfish Immunol. 2018, 76:101–109 Faggio, C.; Tsarpali, V.; Dailianis, S. Mussel digestive gland as a model for assessing xenobiotics: an overview. Sci Total Environ. 2018, 613: 220-229; Aliko, V.; Hajdaraj, G.; Caci, A.; Faggio, C. Copper Induced Lysosomal Membrane Destabilisation in Haemolymph Cells of Mediterranean Green Crab (Carcinus aestuarii, Nardo, 1847) from the Narta Lagoon (Albania). Brazilian Archives of Biology and Technology. 2015, 58 (5): 750-756; Pagano, M.; Porcino, C.; Briglia, M.; Fiorino, E.; Vazzana, M.; Silvestro, S.; Faggio, C. The influence of exposure of cadmium chloride and zinc chloride on haemolymph and digestive gland cells from Mytilus galloprovincialis. Int. J. Environ. Res. 2017, 11(2): 207-216 Vajargah, M.F.; Yalsuyi, Am.; Sattari, M.; Prokic, M.; Faggio, C. Effects of Copper Oxide Nanoparticles (CuO-NPs) on Parturition Time, Survival Rate and Reproductive Success of Guppy Fish, Poecilia reticulate. Journal of Cluster Science 2019. in press 10.1007/s10876-019-01664-y; Savorelli, F.; Manfra, L.; Croppo, M.; Tornambè, A.; Palazzi, D.; Canepa, S.; Trentini, P.L.; Cicero, A.M.; Faggio C. Fitness evaluation of Ruditapes philippinarum exposed to nickel. Biological Trace Element Research 2017, 177(2): 384-393; Torre, A.; Trischitta, F.; Faggio C. Effect of CdCl2 on Regulatory Volume Decrease (RVD) in Mytilus galloprovincialis digestive cells. Toxicology in Vitro. 2013,.27: 1260–1266) Kondera, E.; Ługowska, K.; Sarnowski, P. High affinity of cadmium and copper to head kidney of common carp (Cyprinus carpio). Fish Physiol Biochem. 2014, 40, 9–22, DOI 10.1007/s10695-013-9819-1. Zhang, N.; Zang, S.; Sun, Q. Health risk assessment of heavy metals in the water environment of Zhalong Wetland, China. Ecotoxicology. 2014, 23, 518-526, doi:10.1007/s10646-014-1183-0 Naimo, T.J. A review of the effects of heavy metals on freshwater mussels. Ecotoxicology., 1995, 4, 341-362. Lu, G.; Yang, X.; Li, Z.; Zhao, H.; Wang, C. Contamination by metals and pharmaceuticals in northern Taihu Lake (China) and its relation to integrated biomarker response in fish. Ecotoxicology. 2013, 22, 50-59, doi:10.1007/s10646-012-1002-4 Liu, M.; Yang, Y.; Yun, X.; Zhang, M.; Li, Q.X.; Wang, J. Distribution and ecological assessment of heavy metals in surface sediments of the East Lake, China. Ecotoxicology. 2014, 23, 92-101, doi:10.1007/s10646-013-1154-x Harangi, S.; Baranyai, E.; Fehér, M.; Tóth, N.; Herman, P.; Stündl, L.; Fábián, S.; Tóthmérész, B.; Simon, E. Accumulation of metals in juvenile carp (Cyprinus carpio) exposed to sublethal levels of iron and manganese: survival, body weight and tissue. Biol. Trace Elem. Res. 2017, 177, 187-195, doi:10.1007/s12011-016-0854-5 Hauser-Davisa, R. A.; Bordonb, I. C.A.C.; Oliveirac, T.F.; Ziolli, R.L. Metal bioaccumulation in edible target tissues of mullet (Mugil liza) from a tropical bay in Southeastern Brazil. J Trace Elem in Med Bio. 2016, 36, 38-43, http://dx.doi.org/10.1016/j.jtemb.2016.03.016 Lourenco, H.M.; Afonso, C.; Anacleto, P.; Martins, M.F.; Nunes, M.L.; Nino, A.R. Elemental composition of four farmed fish produced in Portugal. Int. J. Food Sci. Nutr., 2012, 63, 853–859, doi:10.3109/09637486.2012.681632 Martinez, B.; Miranda, J.M.; Nebot, C.; Rodriguez, J.L.; Cepeda, A.; Franco, C.M. Differentiation of farmed and wild turbot (Psetta maxima): Proximate chemical composition, fatty acid profile, trace minerals and antimicrobial resistance of contaminant bacteria. Food Sci. Technol. Int. 2010, 16(5), 435-41, doi:10.1177/1082013210367819 Stanchev, H.; Palazov, A.; Stancheva, M.; Apostolov, A. Determination of the Black Sea area and coastline length using GIS methods and Landsat 7 satellite images. Geo-Eco-Mar. 2011, 17, 27-31 Popescu, I. Directorate general for internal policies, Policy Department B: Structural and cohesion policies, Fisheries in the Black Sea, Note, European Parliament, 2010. Stancheva, M.; Georgieva, S.; Makedonski, L. Polychlorinated biphenyls in fish from Black Sea. Bulgaria. Food Control. 2017, 72, 205-210, doi:10.1016/j.foodcont.2016.05.012 Malakhova, L.; Giragosov, V.; Khanaychenko, A.; Malakhova, T.; Egorov, V.; Smirnov, D. Partitioning and Level of Organochlorine Compounds in the Tissues of the Black Sea Turbot at the South-Western Shelf of Crimea. Turk J Fish Aquat Sci. 2014, 14, 993-1000, doi:10.4194/1303-2712-v14_4_19 Li, X.; Chen, Y.; Cai, L.; Xu,Y.; Yi, S.; Zhu, W.; Mi, H.; Li, J.; Lin, H. Freshness assessment of turbot (Scophthalmus maximus) by Quality Index Method (QIM), biochemical, and proteomic methods. LWT - FOOD SCI TECHNOL. 2017, 78, 172-180, doi:10.1016/j.lwt.2016.12.037 Lyu, D.; Wang, W.; Luan, S.; Hu, Y.; Kong, J. Estimating genetic parameters for growth traits with molecular relatedness in turbot (Scophthalmus maximus, Linnaeus). Aquaculture. 2017, 468, 149-155, doi:10.1016/j.aquaculture.2016.09.049 Li, X.; Chi, L.; Tian, H.; Meng, L.; Zheng, J.; Gao, X.; Liu, Y. Colour preferences of juvenile turbot (Scophthalmus maximus). Physiol. Behav. 2016, 156, 64-70, doi:10.1016/j.physbeh.2016.01.007 Diaz de Cerio, O.; Bilbao, E.; Ruiz, P.; Pardo, B.G.; Martinez, P.; Cajaraville, M.P.; Cancio, I. Hepatic gene transcription profiles in turbot (Scophthalmus maximus) experimentally exposed to heavy fuel oil nº 6 and to styrene. Mar. Environ. Res. 2017, 123, 14-24, doi:10.1016/j.marenvres.2016.11.005 Jia, R.; Han, C.; Lei, J.L.; Liu, B.L.; Huang, B., Huo, H.H.; Yin, S.T. Effects of nitrite exposure on haematological parameters, oxidative stress and apoptosis in juvenile turbot (Scophthalmus maximus). Aquat. Toxicol. 2015, 169, 1-9, doi:10.1016/j.aquatox.2015.09.016 Jia, R.; Liu, B.L.; Han, C.; Huang, B.; Lei, J.L. The physiological performance and immune response of juvenile turbot (Scophthalmus maximus) to nitrite exposure. Comp. Biochem, Physiol. Part C Toxicol. Pharmcol. 2016, 181-182, 40-46, doi:10.1016/j.cbpc.2016.01.002 Aksungur, N.; Aksungur, M.; Akbulut, B.; Kutlu, I. Effects of Stocking Density on Growth Performance, Survival and Food Conversion Ratio of Turbot (Psetta maxima) in the Net Cages on the Southeastern Coast of the Black Sea. Turk J Fish Aquat Sci. 2007, 7, 147-152. Tuzen, M. Determination of heavy metals in fish samples of the middle Black Sea (Turkey) by graphit furnace atomic absorbtion spectrometry. Food Chem. 2003, 80, 119-123, doi:10.1016/S0308-8146(02)00264-9 Tuzen, M. Toxic and essential trace elemental content in fish species from the Black Sea, Turkey. Food Chem. Toxicol. 2009, 47, 1785-1790, doi:10.1016/j.fct.2009.04.029 Bat, L.; Gundogdu, A.; Yardim, O.; Zoral, T.; Culha, S. Heavy metal amounts in zooplankton and some commercial teleost fish from inner harbor of Sinop, Black Sea. Su Urun. Muh. Derg. 2006, 25, 22-27. Das, Y. K. Aksoy, A. Baskaya, R.H. Duyar, A.D. Guvenc, V. Boz,, Heavy metal levels of some marine organisms collected in Samsun and Sinop coasts of Black Sea, in Turkey, Journal of Animal and Veterinary Advances 8(3) (2009)496-499. Nisbet, C.G. Terzi, O. Pilger, N. Sarac, Determination of heavy metal levels in fish samples collected from the Middle Black Sea, Kafkas Univ. Vet. Fak. Derg. 16 (2010)119-125, doi:10.9775/kvfd.2009.982 Ergönül, M.B., Altindağ, A.Heavy metal concentrations in the muscle tissues of seven commercial fish species from Sinop coasts of the Black Sea, Annual Set the Environment Protection, 16 (2014) 34-51. Manthey-Karl, M.; Lehmann, I.; Ostermeyer, U.; Schroder, U. Natural chemical composition of commercial fish species: Characterisation of pangasius, wild and farmed turbot and barramundi. Foods 2016, 5(3), 58, doi:10.3390/foods5030058 Yanchilina A. G., Ryan W. B. F., McManus J.F., Dimitrov P., Dimitrov D., Slavova K, M. Filipova-Marinova, Compilation of geophysical, geochronological, and geochemical evidence indicates a rapid Mediterranean-derived submergence of the Black Sea's shelf and subsequent substantial salinification in the early Holocene, Mar. Geol., 383 (2017)14-34, doi:10.1016/j.margeo.2016.11.001 Black Sea Commission (2007) The Commission of the protection of the Black Sea against pollution, Black Sea Transboundary Diagnostic Analysis. Plavan G., Jitar O., Teodosiu C., Nicoara M., Micu D., Strungaru S.A., Toxic metals in tissues of fishes from the Black Sea and associated human health risk exposure, Environ. Sci. Pollut. Res., 24 (2017) 7776-7787, doi:10.1007/s11356-017-8442-6 Zaitsev Y. P., Alexandrov B. G., Berlinsky N. A., Zenetos A., Europe's biodiversity - biogeographical regions and seas, Seas around Europe The Black Sea - an oxygen-poor sea. European Environment Agency (2002). Jitar O., Teodosiu C., Oros A., Plavan G., Nicoara M., Bioaccumulation of heavy metals in marine organisms from the Romanian sector of the Black Sea, N. Biotechnol., 32 (2014) 369-378, doi:10.1016/j.nbt.2014.11.004 Concalves, C.; Martins, M.; Diniz, M.S.; Costa, M.H.; Cairo, S.; Costa, P.M. May sediment contamination be xenoestrogenic to benthic fish? A case study with Solea senegalensis. Mar Environ Res. 2014, 99, 170-178 Cuevas, N.; Zorita, I.; Costa, P.M.; Larreta, J.; Franco, J. Histopathological baseline levels and confounding factors in common sole (Solea solea) for marine environmental risk assessment. Mar Environ Res. 2015, 110, 162-173, doi:10.1016/j.marenvres.2015.09.002. Samsun, N.; Kalayci, F. Survival rates of black sea turbot (Scophthalmus maeoticus Pallas, 1811) captured by bottom turbot gillnets in different depths and fishing seasons between 1999 and 2004. Turk J Fish Aquat Sci. 2005, 5, 57-62 Whitehead, P.J.P.; Bauchot, M.L.; Hureau, J.C.; Nielsen, J.; Tortonese, E. Fishes of the North-eastern Atlantic and The Mediterranean. UNESCO. 1986, Vol. 3, 1287–1293 https://www.fishbase.se/summary/Scophthalmus-maximus.html Radu G., Radu E., Taxonomically identifier for Black Sea fish species – in romanian language (Determinator al principalelor specii de pesti din Marea Neagra), VIROM Publishing Company, ISBN: 978-973-7895- 33-2, 2008, 504 – 509. Yi Y.J., Zhang S.H., The relationships between fish heavy metal concentrations and fish size in the upper and middle reach of Yangtze River, Procedia Environmental Sciences, 2012, 13, 1699-1707. Merciai R., Guasch H., Kumar A., Sabater S., Trace metal concentration and fish size: Variation among fish species in a Mediterranean river, Ecotoxicology and Environmental Safety, 2014, 107, 154-161. Monsefrad F., Imanpour N., Heidary S., Concentration of heavy and toxic metals Cu, Zn, Cd, Pb and Hg in liver and muscles of Rutilus frisii kutum during spawning season with respect to growth parameters, Iranian Journal of Fisheries Science, 2012, 11 (4), 825-839. Maximov V., Zaharia T., Nicolaev S., Radu G., State of the Romanian Black Sea Turbot (Psetta maxima maoetica) resources. Recherches Marines, 43 (2013) 296-306 Eryilmaz L., Dalyan C., Age, growth, and reproductive biology of turbot, Scophthalmus maximus (Actinopterygii: Pleuronectiformes: Scophthalmidae), from the south-western coasts of Black Sea, Turkey; ACTA ICHTHYOLOGICA ET PISCATORIA (2015) 45 (2): 181–188. Strungaru, S.A.; Nicoara, M.; Jitar, O.; Plavan, G. Influence of urban activity in modifying water parameters, concentration and uptake of heavy metals in Typha latifolia into a river that crosses an industrial city. Journal of Env. Health Sci. & Engineering. 2015, 13:5, 1-15 doi:10.1186/s40201-015-0161-7 Bulska E., Krata A., Kalabun M., Wojciechowski M., On the use of certified reference materials for assuring the quality of results for the determination of mercury in environmental samples, Environmental Science and Pollution Research, 2017, 24 (9), 7889-7897. Jitar, O., Teodosiu C., Oros A., Plavan G., Nicoara M., Bioaccumulation of heavy metals in marine organism from the Romanian sector of the Black Sea, New Biotechnology, 2014, 32, 369-378. Bury N., Walker P.A., Glover C.N., Nutritive metal uptake in teleost fish, The Journal of Experimental Biology, 2003, 206, 11-23. Lall, S.P.; Lewis-McCrea, L.M. Role of nutrients in skeletal metabolism and pathology in fish -An overview. Aquaculture 2007, 267, 3-19, doi:10.1016/j.aquaculture.2007.02.053 FOOD AND AGRICULTURE ORGANIZATION OF THE UNITED NATIONS - UNITED NATIONS DEVELOPMENT PROGRAMME, ADCP/REP/80/11 - Fish Feed Technology, Chapter 7, ISBN 92-5-100901-5, 1980, Wei, Y.; Zhang, J.Y.; Zhang, D.W.; Tu, T.H.; Luo, L.G. Metal concentrations in various fish organs of different fish species from Poyang Lake, China. Ecotoxicol. Environ. Saf. 2014, 104, 182-188, doi:10.1016/j.ecoenv.2014.03.001 Bijvelds, M.J.C.; Velden, V.D.; Kolar, Z.I.; Flik, G. Magnesium transport in freshwater teleosts. J. Exp. Biol. 1998, 201, 1981-1990. Vijayan, D. K.; Jayarani, R.; Singh, D.K.; Chatterjee, N.S.; Mathew, S.; Mohanty, B.P.; Sankar, T.V.; Anandan, R. Comparative studies on nutrient profiling of two deep sea fish (Neoepinnula orientalis and Chlorophthalmus corniger) and brackish water fish (Scatophagus argus). JOBAZ. 2016, 77, 41-48, doi:10.1016/j.jobaz.2016.08.003 Heath, A. Uptake, Accumulation, Biotransformation, and Excretion of Xenobiotics, Water pollution and fish physiology, Library of Congress Cataloging in Publication data, Second Edition, Lewis Publishers, Florida. 1995, 79-86. Simionov I.A., Cristea V., Petrea S.M., Bocioc Sîrbu E., Evaluation of heavy metals concentration dynamics in fish from the Black Sea coastal area: an overview, Environmental Engineering and Management Journal, 2019, 18 (5), 1097-1110. Trifan, A.; Breaban, G.; Sava, D.; Bucur, L.; Toma, C. C.; Miron, A. Heavy metal content in macroalgae from Roumanian Black Sea. Rev. Roum. Chim. 2015, 60, 915-920. El-Moselhy, Kh.M.; Othman, A.I.; El-Azem, H.A.; El-Metwally, M.E.A. Bioaccumulation of heavy metals in some tissues of fish in the Red Sea, Egypt. EJBAS. 2014, 1, 97-105, doi:10.1016/j.ejbas.2014.06.001 Wu, S. M.; Shu, L. H.; Liu, J. H. Anti-oxidative functions of mt2 and smtB mRNA expression in the gills and brain of zebrafish (Danio rerio) upon cadmium exposure. Fish Physiol Biochem. 2016, 42, 1709–1720, DOI 10.1007/s10695-016-0251-1 Yuan, S. S.; Lv, Z. M.; Zhu, A.Y.; Zheng, J. L.; Wu, C. W. Negative effect of chronic cadmium exposure on growth, histology, ultrastructure, antioxidant and innate immune responses in the liver of zebrafish: Preventive role of blue light emitting diodes. Ecotoxicol. Environ. Saf. 2017, 139, 18-26, doi:10.1016/j.ecoenv.2017.01.021

Distinguished Editorial Border of Journal of Marine Science and Engineering,

To Reviewer # 2,

            First of all, I would like to thank you for all your effort in reviewing this paper and also, to present apologies for our previous not consistent response for your 1st report.   

            Reviewer # 2 specific comments:

Line 21 – Make a brief state of the art of your manuscript, outlining current research and why your study is important.

Answer: I have integrated into the previous introduction the following paragraphs:

As the Black Sea is considered a semi-enclosed sea and has positive freshwater balance, mainly due to high river inputs, it is exposed to heavy metals contamination, especially in coastal area. Fish are considered top consumers in all aquatic systems and therefore, they are continuously exposed to contaminants, situation which involves tissues bioaccumulation especially in case of demersal species, of both essential and non-essential heavy metals. Thus, as elements have the tendency to accumulate in specific hot spots, it is recommended the use of demersal fish species as bio-indicators, in order to evaluate the heavy metals pollution. Thus, analyzing different fish tissues ensures scanning of aquatic environment, considering fish mobility and capacity to continuously accumulate xenobiotics.

A series of demersal fish species as turbot (Psetta maxima maeotica) and whiting (Merlangius merlangus) can be used as bioindicators in order to evaluate the heavy metals pollution in certain coastal areas, since they are species that do not migrate on long, transboundary distances [25]. Also, there species are involved only in local migrations within the coastal areas for spawning, feeding and wintering [25].

Therefore, the analysis of demersal fish catches in terms of heavy metals concentrations can represent an important instrument in order to evaluate the coastal areas pollution, especially after wintering period, when fish stock migrate closer to shore, at water depths between 20-40m, for spawning.

Among Black Sea demersal fish species, turbot presents interest in research for the biomonitoring of polychlorinated biphenyls [26, 27]; food science preservation – freshness assessment of turbot by using different biochemical and proteomics methods [28]; molecular genetics [29]; ecology and behaviour [30]; biomarker in oil pollution [31]; nitrite toxicity [32, 33]; aquaculture [34]; heavy metal biomonitoring [35-40]. Several research papers conducted by Ergönül and Altindağ, 2014 [40]; Nisbet et al., 2010 [39]; Das et al., 2009 [38]; Bat et al., 2006 [37]; Tuzen 2003, 2009 [26, 36] studied the concentration of potentially toxic metals and metalloids (Zn, Fe, Cu, Ni, Mn, Cr, Cd, Pb, As) in the Black Sea turbot muscle tissue captured in Turkish territorial marine waters. Other scientific articles realised by Manthey-Karl et al., 2016 [41], Martinez et al., 2010 [23] and Lourenco et al., 2012 [22] studied the concentration of essential macro-elements (Ca, Mg, Na, K), micro-elements (Zn, Fe, Cu, Ni, Mn) and potentially toxic metals (Cd, Pb, As) in the muscle tissue of wild turbot captured in Atlantic Ocean, respectively, in farmed turbot from Portugal. However, the studies focused on essential, non-essential elements and toxic metals in the Black Sea turbot organism are incomplete since only potential toxic micro-elements and trace-elements were determined The macro-elements are important to be determined in order to identify their capacity of influence micro and trace-elements bioaccumulation. Also, those studies can contribute to a better evaluation of certain Black Sea coastal areas, considered under a continuous anthropogenic pressure.

This study investigates the influence of turbot gender, catched in the Romanian Black Sea Coast, in bioconcentration of both essential (Ca, Cu, Fe, K, Mg, Mn, Na, Zn) and nonessential elements (As, Cd, Ni) in various tissues (gills - Gi, stomach - St, intestine - In, liver - Lv, spleen - Sp, gonads - Go, muscles - Mu, caudal fin - Cf), including stomach and intestine content. This evaluation will also contribute to an upgraded characterization of Romanian Black Sea Coast, in terms of heavy metals pollution.

Line 23 – What area? River, catchment, country?

Answer: This study investigates the influence of gender in bioconcentration of essential and nonessential elements in different parts of Black Sea turbot (Psetta maxima maeotica) body, from an area considered under high anthropogenic pressure (the Constanta city Black Sea Coastal Area - Romania).

Line 23-24 – “They were measured”. Who was measured??

Answer: A number of 13 elements (Ca, Mg, Na, K, Fe, Zn, Mn, Cu, Ni, Cr, As, Pb and Cd) were measured in various sample types: muscle, stomach, stomach content, intestine, intestine content, gonads, liver, spleen, gills and caudal fin.

Line 26 – How many adults? How many per group? Line 27 – “and they were prepared a high number of samples (1200)”. English language needs to be reviewed by a native speaker: the fish were prepared a high number of samples? This 1200 refers to the name of total samples considering the two groups? Line 28 – Do not use acronyms for these techniques upon first mention.

Answer for aforementioned comments: Turbot adults (4-5 years old) were separated, according to their gender, in two groups (20 males, respectively 20 females) and a high total number of samples (1200 from both groups) were prepared and analysed, in triplicate, with Flame Atomic Absorption Spectrometry and high-resolution continuum source atomic absorption spectrometry with graphite furnace techniques.

Line 28-29 – This seems quite simplistic. You should previously refer what was the purpose of your analysis, of your study.

Answer: The results were statistical analysed in order to emphasize the bioconcentration of the determined elements in different tissues of wild turbot males vs. females and also, to contribute to an upgraded characterization of Romanian Black Sea Coast – Constanta city, in terms of heavy metals pollution.

Line 30-33 – Why were only 2 elements (here the full name, but previously only the acronym: please homogenize) selected out those 13? Again, authors need to clearly point the context and goals of their research. Line 33 – How does this related to your goal outlined on line 21? i.e. what is the influence of sex gender in bioconcentration of essential and nonessential elements in turbot? This needs to be clearer.

Answer for aforementioned comments: The Mg and Zn have different roles in gonads of males and females, as they were the only elements with completely different patterns between the analyzed groups of specimens. The concentrations of studied elements in muscle were not similar with data provided by literature, suggesting that chemistry of the habitat and food plays a major role in the availability of the metals in the body of analyzed fish species. The gender influenced the bioaccumulation process of all analyzed elements in most tissues since turbot male specimens accumulated higher concentration of metals compared to females.  The highest bioaccumulation capacity in terms of Ca, Mg, Na, Ni, As, Zn and Cd was registered in caudal fin, liver and intestine tissues. Also, other elements such as K, Fe, Cu and Mn had the highest bioaccumulation in muscle, spleen, liver and gills tissues.

Line 40 – Could you specify these measures?

Answer: On the other hand, nonessential metals, such as toxic metals resulted in the water from fine suspended solids in the water, may alter the feeding rate of fish and determine a reduction in the metabolic efficiency [3, 4]. Sublethal effects manifested when fish are exposed to acute metals concentrations include damage of sensory organs and receptors, which leads to impairment of sensorial perception (namely olfaction) [4] According to Gati et al., 2016, sediment samples with a higher value than 4 of the probable effect concentration quotient (PEC-Q) are more likely to have acute or toxic effects on benthic organisms [3]. In this way, integrated measures for toxicological risk ranking in fish communities were developed [5].

Line 65 – Remove: Line 66 – What do you mean by food science preservation?

Answer for aforementioned comments:  Among Black Sea demersal fish species, turbot presents interest in research for the biomonitoring of polychlorinated biphenyls [26, 27]; food science preservation – freshness assessment of turbot by using different biochemical and proteomics methods [28]; molecular genetics [29]; ecology and behaviour [30]; biomarker in oil pollution [31]; nitrite toxicity [32, 33]; aquaculture [34]; heavy metal biomonitoring [35-40].

Line 68 – Could you specify these elements within each group and outline their importance? Line 69 – Why are incomplete and above all, why they are important both from the theoretical and practical point of view?

Answer for aforementioned comments: Several research papers conducted by Ergönül and Altindağ, 2014 [40]; Nisbet et al., 2010 [39]; Das et al., 2009 [38]; Bat et al., 2006 [37]; Tuzen 2003, 2009 [26, 36] studied the concentration of potentially toxic metals and metalloids (Zn, Fe, Cu, Ni, Mn, Cr, Cd, Pb, As) in the Black Sea turbot muscle tissue captured in Turkish territorial marine waters. Other scientific articles realised by Manthey-Karl et al., 2016 [41], Martinez et al., 2010 [23] and Lourenco et al., 2012 [22] studied the concentration of essential macro-elements (Ca, Mg, Na, K), micro-elements (Zn, Fe, Cu, Ni, Mn) and potentially toxic metals (Cd, Pb, As) in the muscle tissue of wild turbot captured in Atlantic Ocean, respectively, in farmed turbot from Portugal. However, the aforementioned studies focused on essential, non-essential elements and toxic metals in the Black Sea turbot organism are incomplete since only potential toxic micro-elements and trace-elements were determined. The macro-elements are important to be determined in order to identify their capacity of influence micro and trace-elements bioaccumulation. Also, those studies can contribute to a better evaluation of certain Black Sea coastal areas, considered under a continuous anthropogenic pressure.

Line 70-72 – Again, you should previously point out the role of essential and non-essential elements.

Answer: This study investigates the influence of turbot gender, catched in the Romanian Black Sea Coast, in bioconcentration of both essential (Ca, Cu, Fe, K, Mg, Mn, Na, Zn) and nonessential elements (As, Cd, Ni) in various tissues (gills - Gi, stomach - St, intestine - In, liver - Lv, spleen - Sp, gonads - Go, muscles - Mu, caudal fin - Cf), including stomach and intestine content. This evaluation will also contribute to an upgraded characterization of Romanian Black Sea Coast, in terms of heavy metals pollution.

Line 86 – Which biological material? The turbot? how many? weight? Line 90 – It is therefore supposed that the biological material (fish?) are contaminated by such human pressures. Line 91 – I think this line can be removed (don´t need to outline software to produce the maps) Line 95 – This line can be removed. Line 96-97 –Where were they captured and how? Were all form the same site? What is their size-range? Line 99 – “measured and weighted”. To the nearest cm and g? Line 99-101 – This is a result and should not be place in this section (Material and methods). Line 100- “for all biometric variables”. This is not true for head length (cm). Line 101 – test instead of Test. Line 104- Give complete captions: indicate study area, goal,..

Answer for aforementioned comments:  The area of study was chosen since it is the most intensely populated from the Romanian coastline and also, the main anthropogenic activities (tourism, municipal wastewater effluents, industry, cargo shipping and oil refinery) are concentrated here [43, 46]. The map of the Romanian Black Sea targeted coastal area is presented in figure 1.

Since fish stocks are affected by high anthropogenic pressure, the biological material in present study is represented by a number of 40 Black Sea turbot specimens. The biological material was selected from a larger number of turbot specimens purchased fresh from the fish market located in Constanta City and surrounding area, in April 2016 (Fig. 1). All turbot specimens were catched in the fishing areas along the coastline of Constanta City, by using bottom (turbot) gill nets with a minimum mesh size of 180 mm. Flatfish are considered to be suitable bioindicators of pollution due to their ecological importance, sensitivity and close contact to sediment substrates [47].

The studied material was taxonomically identified as Psetta maxima maeotica Linnaeus, 1758, by using both an online fish taxonomically identification database [51] and a taxonomic identifier for Black Sea fish species [52]. The gender identification was made in situ since we had the permission to do a small abdominal incision, with plastic laboratory instruments, for each fish, to determine the sex gender, in order assure the targeted sampling number for this study (20 males and 20 females). This was also useful for the identification and selection of the fish specimens that had stomach content. A number of 40 turbot specimens (20 males and 20 females) were placed in polyethylene bags, stored on ice and transported to the laboratory, where they were frozen at -20°C, until the samples were prepared and analysed. Before the analysis, the specimens were dissected by using plastic laboratory instruments, for avoiding samples contamination. Each individual was biometric measured and individual biomass was determined before frozen at -20°C since, according to different authors, the fish size can influence the bioaccumulation process [53-55]. The results are presented in Table 1. In this study, the males group had significant (*p<0.05) higher values than the female group for all biometric variables according to t-test results. – it was removed in the results and discussions section, together with table 1.  Based on biometric measurements, all specimens were considered as part of a cohort of 4-5 years old adults, capable for reproduction, according to Maximov et al. [56], that collected samples, in its scientific study, from similar area – Black Sea Coast - Constanta City. The age of turbot specimens was determined according to Eryilmaz and Dalyan method [57]

Table 1. Biometric measurements and individual biomass of the studied fish (mean±SD).  – head length average values were reversed because they were wrong placed previously

Gender

Biometric measurement

Female

Male

Total length (cm)

44.60±1.86

48.3±1.55

Maximum width (cm)

32.6±2.15

35.1±0.28

Individual biomass (kg)

1.83±0.29

2.32±0.12

Distance between eyes (cm)

1.4±0.05

1.5±0.02

Caudal fin length (cm)

8.24±0.05

10.18±0.3

Head length (cm)

8.74±0.67

10.21±0.49

Head maximum width (cm)

8.9±0.02

12.56±0.56

Line 105 – What was the quantity of tissue removed in each case?

Answer: After dissection, the following sample types were collected (1 g each) for analysis: muscle, stomach, stomach content, intestine, intestine content, gonads, liver, spleen, gills and caudal fin. All the analysed specimens had over 70% stomach content, reported to the entire stomach capacity.

Line 116 – 30 samples x 40 fish? If so, please include.

Answer: The total amount of prepared and analysed samples in this study was equal to 1200 (10 samples per each fish specimen, in triplicate).

Line 118-119 – equipment brand and model should be in parenthesis.

Answer: For the metal quantification analyses two methods were used: flame atomic absorption spectrometer FL-AAS(GBC Avanta, Australia), for calcium (Ca), magnesium (Mg), sodium (Na), potassium (K), iron (Fe), zinc (Zn) and high resolution continuum source atomic absorption spectrometer with graphite furnace HR-CS GF-AAS (equipment ContrAA 600-Analytik Jena, Germany), for copper (Cu), manganese (Mn), nickel (Ni), arsenic (As), cadmium (Cd), lead (Pb), chromium (Cr).

Line 122 – Why do you need to validate the methods? Line 123 – “ERM-BB422”. What does this mean?

Answer:  Certified reference materials (CRMs) are recognized to be an essential tool in assuring the accuracy and establishing the traceability of the results of measurements [59]. Therefore, the reference material for fish muscle (ERM- BB422 type certified reference material) was analysed, certified by the Joint Research Center Institute for Reference Materials and Measurements.

Line 124 – Are these other fish other than the 40? replace “replicas” by “replicates”. Line 125 – Again, Results should not be presented in the Material and Methods section. Table 2 – What do you mean by confidence? Is it the standard deviation (SD) as you have on Table 1? Be consistent!

Answer for aforementioned comments:  The aforementioned reference material was prepared in 6 replicates, following the same protocol as a normal sample [44], in order to eliminate any determination errors and the results obtained were presented in Table 2. The certificate of analysis contained the following elements: As, Cd, Cu, Fe, Hg, I, Mn, Se, Zn, Ca, Cl, K, Mg, Na.  – in our opinion, this is part of material and methods section since it is a step that assures the accuracy and establish the traceability of the results of measurements.

Table 2. Method for assuring the accuracy and establishing the traceability of the measurements results, based on the use of CRM (µg g-1 dry weight). – I have change confidence with SD

Measured

element

Certified value mean±SD

 (µg g-1 )

Measured value mean±SD

 (µg g-1 )

Analytical method

Number of replicates

As

12.7±0.7

11.3±0.4

HR-CS GF-AAS

6

Ca

342

340±6

FL-AAS

6

Cd

0.0075±0.0018

0.0077±0.002

HR-CS GF-AAS

6

Cu

1.67±0.16

1.62±0.27

HR-CS GF-AAS

6

Fe

9.4

9.5±0.3

FL-AAS

6

K

21400

21381±21

FL-AAS

6

Mg

1370

1372±7

FL-AAS

6

Mn

0.368±0.028

0.361±0.034

HR-CS GF-AAS

6

Na

2800

2771±27

FL-AAS

6

Zn

16±1.1

15±1.7

FL-AAS

6

Line 129 – Did you also test your data for homoscedasticity? What kind of data was tested? Line 134 – Was it not the same as you did with the ANOVA?

Answer for aforementioned comments: We did not do homoscedasticity tests because we considered that ANOVA, applied for analyzing data, is based on the normality assumptions of error component (random component or residual component). Accordingly, it is not necessary to the residuals for normality and homogeneity of variance.

First, the Levene’s homogeneity variance test andthe Kolmogorov-Smirnov normality test were performed to study the data distribution in both experimental groups (males and females). The result of each test proved that the data had a normal distribution. After this step they were performed one-way ANOVA for analysis of the variance significance for each element concentration between tissues that were sampled from both genders, followed by multiple comparisons Tukey HSD test, in order to determined which of the means are different. The t-Test was performed to observe any significant differences between females and males regarding all the concentrations of studied elements for each sample type. All statistical analysis was carried out using OriginPro v.9.3. (2016) software (OriginLab Corporation, USA). All the data for metal concentration were presented in this study as average±SD. The mean values registered for different metals concentrations in the muscle tissue were centralised and compared to data reported in other studies, which sampled muscle tissue from aquaculture and wild turbot.

Line 137 – Could you provide a reference and a brief explanation for this index?

Answer: The bioconcentration factor was calculated for dietary exposure according to the formula (1) [60].

Line 139 – Any evidence to support this?

Answer: The main route of absorption of the elements in the body was considered to be the food and sediments that were ingested, according to Bury et al. who pointed out that the diet is the main source of essential metals in the fish body [61].

Line 140-141 – This is not clear and needs to be better explained. Do you have a reference/study to support this index?

Answer: The value of this factor indicates the capacity of a specific biological sample to accumulate a certain part from the total element input in the fish body. The index was used also in other similar studies [61].

Line 144 – LOD?

Answer: The Pb and Cr were below the detection limit of quantification (LOQ) for the calibration curve method (LOQ Pb-0.032 µg L-1, LOQ Cr-0.3 µg L-1) in all analyzed samples, excepting for the stomach samples where was measured Cr.

Table 3 – Use the same number of decimals for each pair of entries (i.e. males and females). Line 161-162 – It seems there is some speculation here; there is no references/previous studies to support this.

Answer: This could be related to turbot diet and its mobility in various areas with different elements concentration, where the food resources are located [25].

2 (lines 175-179) – What’s the name of this analysis? I do not see it adds much to the results on Table 3.

Answer: Figure 2. The representation of elements affinity and variation in the body of male and female specimens of Black Sea turbot, based on the registered results (µg g-1 fresh weight- f.w.). (one-way ANOVA analysis was applied for males and females in order to determine the variation of each studied element ;*p<0.05).

The figure 2 emphasizes the variations and affinity of elements in the body of male and female specimens of Black Sea turbot, based on the registered results recorded from the analysed sample tissues. It must be noted that the studied males were bigger than the females. This can be one of the explanations and the role of these elements in gonads.

Line 179-180 – This sentence is unclear. The results of the ANOVA are the test statistic and significance value and not “physiological demand and bioconcentration affinity the turbot body.” Table 4 – This table is redundant to Table 3, as it a simple ordination (from higher to lower) of the concentrations in each organ. What does the table show indeed’ It says only “The results of multiple comparisons…”

Answer: We have deleted both table 4 and the ,, The results of the ANOVA are the test statistic and significance value and not “physiological demand and bioconcentration affinity the turbot body.”

Line 186 – “In this case the stomach content was used to measure it.”. Could you clarify? Line 187 – “The analysis of the bioaccumulation from sediments was considered to be not entirely correct”. This is a concern if you are doubting of the quality of your analyses. Line 190 – This should be said earlier. Same on line 193 Line 191 – Again this is pure speculation and should be removed. Same on line 222 Line 194 – What evidence from the Table show it was different? Did you make in statistical test? Line 195-196 – I do not understand what the authors mean with this, but this was not said previously In Material and methods.

Answer for aforementioned comments:  The bioconcentration factor of the essential and nonessential elements in the Black Sea turbot was calculated from the dietary intake. In this case, the dietary intake is represented by the stomach content. It must be highlighted that if bioaccumulation analysis is conducted in relation to sediments, the results could be not entirely correct since Black Sea turbot specimens are mobile benthic organisms. The chemistry of Black Sea sediments differs from an area to another [46] and the location where the captured specimens lived cannot be identified exactly since turbot is capable to travel significant distances in search for food [25]. Thus, in the present study it has been chosen to calculate BCF of non-essential and essential elements, for all analysed tissues, in relation to stomach content samples. In table 4 were presented the values of bioconcentration factor calculated from dietary exposure, both for male and female specimens. Each element manifests a bioconcentration propensity for a specific organ. The capacity trend of bioconcentration was same for males and females, excepting the gonads where statistically different results, (p<0.05) – ANOVA, followed by Tukey HSD test, were recorded. The exposure history for each specimen and the area from which they come from are both unknown.

Line 200 – What papers? Specify. Line 203 – “Most of the studies were concerned on this subject because turbot meat has economic value”. You said previously that there were few studies (see abstract and Intro).

Answer for aforementioned comments:  The role of essential and nonessential elements on various fish species was described by scientific papers [62, 63]. Not all of the elements were able to be studied, situation that generates incomplete answers. Our results were compared with studies regarding the essential and nonessential elements from muscle samples of farmed turbot [22] or wild turbot from Atlantic Ocean [23, 41]. The scientific studies on turbot were driven by high economic value of turbot meat. Thus, not all the elements were investigated, in detail.

Line 241 – Both. Which ones? You only mention one, Manthey-Karl et al. [41].

Answer: The Na is known to be involved in important physiological activities, such as sodium channels, and in the osmotic pressure regulation [66]. The blood of a marine fish contains salt at a concentration of one-third of that of full-strength seawater, thus, fish drink a specific amount of water (0.5% body weight/hour) to regulate the osmotic pressure [67]. The mean Na values in the turbot muscles are similar with the results obtained by Manthey-Karl et al. [41], explained by the fact that both studies (present and aforementioned) analysed samples fromwild turbot specimens. The Na pathway in the body of analysed turbot specimens did not significantly differ between genders, even if males concentrated more Na in the caudal fin, stomach content and gills, compare to females. The main route of NaCl uptake is the intestine, while gills represent the main excretion route [67].

Line 250 – This is too vague and requires further explanations/support.

Answer: Zinc mean concentrations in our turbot muscles samples were similar to the results obtained by Ergönül and Altindağ [40] for wild turbot from the Black Sea. Twice higher Zn concentrations were registered in turbot muscle, compared to the results recorded by Lourenco et al [22] and Manthey-Karl et al [41]. Also, the Zn values of Bat et al. [37] and Tuzen [36] (which also analysed Black Sea turbot specimens) were two, respectively three times higher in the turbot muscles, compared to the concentrations registered in present study. This may be due to the applied analysis methodology (the aforementioned studies did not use CRM for assuring the accuracy and establishing the traceability of the measurements results) or sampling area (the Black Sea Turkish coast has been observed to be more polluted in terms of heavy metals, compared to the the Romanian coast [68].

Line 259 – The units in your results are different from the ones of Lourenço et al.

Answer: According to Lourenco et al. [22], Fe was present in concentrations of approximately 5.0 µg g-1 in marine fish species, which was similar to present study results. – Lourenço et al. used mg kg -1, that is equivalent with µg g-1. Therefore, I have changed only the units and the concentration value remains the same. 

Line 293- Previously you call essential and non-essential elements, now you call them macro- and microelements. Be consistent.

Answer: Essential and non-essential elements concentrations in the turbot muscle (mean±SD) µg g-1, f.w.-fresh weight, d.w.-dry weight.

Table 6 – This analysis was not referred on Material and methods as well as its purpose. Please include it.

Answer: Table 5 presents a comparative study between mean concentrations of elements recorded in present study and the concentrations reported, in their research, by other authors. The comparative study revealed a high variability related to essential and non-essential elements concentrations in turbot specimens from the Black Sea and other turbot subspecies, from different parts of the world. So far, this study covers the lack of published data in present area of interest, between the years 2006-2016.

Line 298 – “were very fast concentrated”. How do you know it was fast? Line 300 – Where is the map shown? Line 301-302 – again there is speculation here that should be removed. Line 303-304 – Any reference to support this? Line 304-306 - This seems logical and hence, the authors are now showing any novelty here. Line 306 – Again, these isolation and speculative sentences with no support, should be removed.

Answer for aforementioned comments:  The bioconcentration of essential metals in specific organs was related with their biological function. As a conclusion, it can be stated that the highest bioaccumulation capacity in terms of Ca, Mg, Na, Ni, As, Zn and Cd was registered in caudal fin, liver and intestine tissues. Also, other elements such as K, Fe, Cu and Mn had the highest bioaccumulation in muscle, spleen, liver and gills tissues. The turbot male specimens accumulated higher concentration of metals compared to females. However, this can be also influence in a small percentage, by the higher body size of males, compared to females, since they have approximately the same age. The concentrations of toxic metals in Black Sea turbot from this study were lower in the muscle samples compared with the studies conducted in Turkey suggesting that the anthropogenic activity in the studied area had not a major impact on the habitat contamination. These results can be applied in food development of turbot from aquaculture for essentials and non-essential elements balance. Also, the results reflect the status of Black Sea Coast – Constanta city local environmental condition, since they do not migrate on long, transboundary distances.

The underline text is meant to serve as additional information for you and those comments are not included in the present paper. The rest of the answers, that are not underlined are included in the manuscript.

Also, we have revised the English language of the present paper.

Thank you again for your understanding and we hope we have fulfilled all the specified notes, comments and suggestions from your review report.

Song, Y. F.; Luo, Z.; Huang, C.; Liu, X.; Pan, Y. X.; Chen, Q. L. Effects of calcium and copper exposure on lipogenic metabolism, metal element compositions and histology in Synechogobius hasta. Fish Physiol Biochem. 2013, 39, 1641–1656, DOI 10.1007/s10695-013-9816-4 Santos, I.; Diniz, M.S.; Carvalho, M.L.; Santos, J.P. Assessment of essential elements and heavy metals content on Mytilus galloprovincialis from River Tagus estuary. Biol. Trace. Elem. Res. 2014, 159, 233-240, doi:10.1007/s12011-014-9974-y Gati, G.; Pop, C.; Brudasca, F.; Gurzau, A.E.; Spinu, M. The ecological risk of heavy metals in sediment from the Danube Delta. Ecotoxicology. 2016, 25, 688-696, doi:10.1007/s10646-016-1627-9 Castro, B.B.; Sobral, O.; Guilhermino, L.; Ribeiro, R. An in-situ bioassay integrating individual and biochemical responses using small fish species. Ecotoxicology. 2004, 13, 667-681, doi:10.1007/s10646-003-4427-y Hartwell, S.I.; Dawson, C.E.; Durell, E.Q.; Alden, R.W.; Adolphson, P.C.; Wright, D.A.; Coelho, G.M.; Magee, J.A. Integrated measures of ambient toxicity and fish community diversity in Chesapeake Bay tributaries. Ecotoxicology. 1998, 7, 19-35 Lino, A.S.; Galvão, P.M.A.; Longo, R.T.L.; Azevedo-Silva, C.E.; Dorneles, P.R.; Torres, J.P.M.; Malm, O. Metal bioaccumulation in consumed marine bivalves in Southeast Brazilian coast. J Trace Elem in Med Bio. 2016, 34, 50-55, http://dx.doi.org/10.1016/j.jtemb.2015.12.004 Capillo, G.; Silvestro, S.; Sanfilippo, M.; Fiorino, E.; Giangrosso G.; Ferrantelli V.; Vazzana I.; Faggio C. Assessment of electrolytes and metals profile of the Faro Lake (Capo Peloro Lagoon, Sicily, Italy) and its impact on Mytilus galloprovincialis. Chemestry & Biodiversity, 2018.doi 10.1002/cbdv.201800044; Aliko, V.; Qirjo, M.; Sula, E.; Morina ,V.; Faggio, C. Antioxidant defense system, immune response and erythron profile modulation in Gold fish, Carassius auratus, after acute manganese treatment. Fish Shellfish Immunol. 2018, 76:101–109 Faggio, C.; Tsarpali, V.; Dailianis, S. Mussel digestive gland as a model for assessing xenobiotics: an overview. Sci Total Environ. 2018, 613: 220-229; Aliko, V.; Hajdaraj, G.; Caci, A.; Faggio, C. Copper Induced Lysosomal Membrane Destabilisation in Haemolymph Cells of Mediterranean Green Crab (Carcinus aestuarii, Nardo, 1847) from the Narta Lagoon (Albania). Brazilian Archives of Biology and Technology. 2015, 58 (5): 750-756; Pagano, M.; Porcino, C.; Briglia, M.; Fiorino, E.; Vazzana, M.; Silvestro, S.; Faggio, C. The influence of exposure of cadmium chloride and zinc chloride on haemolymph and digestive gland cells from Mytilus galloprovincialis. Int. J. Environ. Res. 2017, 11(2): 207-216 Vajargah, M.F.; Yalsuyi, Am.; Sattari, M.; Prokic, M.; Faggio, C. Effects of Copper Oxide Nanoparticles (CuO-NPs) on Parturition Time, Survival Rate and Reproductive Success of Guppy Fish, Poecilia reticulate. Journal of Cluster Science 2019. in press 10.1007/s10876-019-01664-y; Savorelli, F.; Manfra, L.; Croppo, M.; Tornambè, A.; Palazzi, D.; Canepa, S.; Trentini, P.L.; Cicero, A.M.; Faggio C. Fitness evaluation of Ruditapes philippinarum exposed to nickel. Biological Trace Element Research 2017, 177(2): 384-393; Torre, A.; Trischitta, F.; Faggio C. Effect of CdCl2 on Regulatory Volume Decrease (RVD) in Mytilus galloprovincialis digestive cells. Toxicology in Vitro. 2013,.27: 1260–1266) Kondera, E.; Ługowska, K.; Sarnowski, P. High affinity of cadmium and copper to head kidney of common carp (Cyprinus carpio). Fish Physiol Biochem. 2014, 40, 9–22, DOI 10.1007/s10695-013-9819-1. Zhang, N.; Zang, S.; Sun, Q. Health risk assessment of heavy metals in the water environment of Zhalong Wetland, China. Ecotoxicology. 2014, 23, 518-526, doi:10.1007/s10646-014-1183-0 Naimo, T.J. A review of the effects of heavy metals on freshwater mussels. Ecotoxicology., 1995, 4, 341-362. Lu, G.; Yang, X.; Li, Z.; Zhao, H.; Wang, C. Contamination by metals and pharmaceuticals in northern Taihu Lake (China) and its relation to integrated biomarker response in fish. Ecotoxicology. 2013, 22, 50-59, doi:10.1007/s10646-012-1002-4 Liu, M.; Yang, Y.; Yun, X.; Zhang, M.; Li, Q.X.; Wang, J. Distribution and ecological assessment of heavy metals in surface sediments of the East Lake, China. Ecotoxicology. 2014, 23, 92-101, doi:10.1007/s10646-013-1154-x Harangi, S.; Baranyai, E.; Fehér, M.; Tóth, N.; Herman, P.; Stündl, L.; Fábián, S.; Tóthmérész, B.; Simon, E. Accumulation of metals in juvenile carp (Cyprinus carpio) exposed to sublethal levels of iron and manganese: survival, body weight and tissue. Biol. Trace Elem. Res. 2017, 177, 187-195, doi:10.1007/s12011-016-0854-5 Hauser-Davisa, R. A.; Bordonb, I. C.A.C.; Oliveirac, T.F.; Ziolli, R.L. Metal bioaccumulation in edible target tissues of mullet (Mugil liza) from a tropical bay in Southeastern Brazil. J Trace Elem in Med Bio. 2016, 36, 38-43, http://dx.doi.org/10.1016/j.jtemb.2016.03.016 Lourenco, H.M.; Afonso, C.; Anacleto, P.; Martins, M.F.; Nunes, M.L.; Nino, A.R. Elemental composition of four farmed fish produced in Portugal. Int. J. Food Sci. Nutr., 2012, 63, 853–859, doi:10.3109/09637486.2012.681632 Martinez, B.; Miranda, J.M.; Nebot, C.; Rodriguez, J.L.; Cepeda, A.; Franco, C.M. Differentiation of farmed and wild turbot (Psetta maxima): Proximate chemical composition, fatty acid profile, trace minerals and antimicrobial resistance of contaminant bacteria. Food Sci. Technol. Int. 2010, 16(5), 435-41, doi:10.1177/1082013210367819 Stanchev, H.; Palazov, A.; Stancheva, M.; Apostolov, A. Determination of the Black Sea area and coastline length using GIS methods and Landsat 7 satellite images. Geo-Eco-Mar. 2011, 17, 27-31 Popescu, I. Directorate general for internal policies, Policy Department B: Structural and cohesion policies, Fisheries in the Black Sea, Note, European Parliament, 2010. Stancheva, M.; Georgieva, S.; Makedonski, L. Polychlorinated biphenyls in fish from Black Sea. Bulgaria. Food Control. 2017, 72, 205-210, doi:10.1016/j.foodcont.2016.05.012 Malakhova, L.; Giragosov, V.; Khanaychenko, A.; Malakhova, T.; Egorov, V.; Smirnov, D. Partitioning and Level of Organochlorine Compounds in the Tissues of the Black Sea Turbot at the South-Western Shelf of Crimea. Turk J Fish Aquat Sci. 2014, 14, 993-1000, doi:10.4194/1303-2712-v14_4_19 Li, X.; Chen, Y.; Cai, L.; Xu,Y.; Yi, S.; Zhu, W.; Mi, H.; Li, J.; Lin, H. Freshness assessment of turbot (Scophthalmus maximus) by Quality Index Method (QIM), biochemical, and proteomic methods. LWT - FOOD SCI TECHNOL. 2017, 78, 172-180, doi:10.1016/j.lwt.2016.12.037 Lyu, D.; Wang, W.; Luan, S.; Hu, Y.; Kong, J. Estimating genetic parameters for growth traits with molecular relatedness in turbot (Scophthalmus maximus, Linnaeus). Aquaculture. 2017, 468, 149-155, doi:10.1016/j.aquaculture.2016.09.049 Li, X.; Chi, L.; Tian, H.; Meng, L.; Zheng, J.; Gao, X.; Liu, Y. Colour preferences of juvenile turbot (Scophthalmus maximus). Physiol. Behav. 2016, 156, 64-70, doi:10.1016/j.physbeh.2016.01.007 Diaz de Cerio, O.; Bilbao, E.; Ruiz, P.; Pardo, B.G.; Martinez, P.; Cajaraville, M.P.; Cancio, I. Hepatic gene transcription profiles in turbot (Scophthalmus maximus) experimentally exposed to heavy fuel oil nº 6 and to styrene. Mar. Environ. Res. 2017, 123, 14-24, doi:10.1016/j.marenvres.2016.11.005 Jia, R.; Han, C.; Lei, J.L.; Liu, B.L.; Huang, B., Huo, H.H.; Yin, S.T. Effects of nitrite exposure on haematological parameters, oxidative stress and apoptosis in juvenile turbot (Scophthalmus maximus). Aquat. Toxicol. 2015, 169, 1-9, doi:10.1016/j.aquatox.2015.09.016 Jia, R.; Liu, B.L.; Han, C.; Huang, B.; Lei, J.L. The physiological performance and immune response of juvenile turbot (Scophthalmus maximus) to nitrite exposure. Comp. Biochem, Physiol. Part C Toxicol. Pharmcol. 2016, 181-182, 40-46, doi:10.1016/j.cbpc.2016.01.002 Aksungur, N.; Aksungur, M.; Akbulut, B.; Kutlu, I. Effects of Stocking Density on Growth Performance, Survival and Food Conversion Ratio of Turbot (Psetta maxima) in the Net Cages on the Southeastern Coast of the Black Sea. Turk J Fish Aquat Sci. 2007, 7, 147-152. Tuzen, M. Determination of heavy metals in fish samples of the middle Black Sea (Turkey) by graphit furnace atomic absorbtion spectrometry. Food Chem. 2003, 80, 119-123, doi:10.1016/S0308-8146(02)00264-9 Tuzen, M. Toxic and essential trace elemental content in fish species from the Black Sea, Turkey. Food Chem. Toxicol. 2009, 47, 1785-1790, doi:10.1016/j.fct.2009.04.029 Bat, L.; Gundogdu, A.; Yardim, O.; Zoral, T.; Culha, S. Heavy metal amounts in zooplankton and some commercial teleost fish from inner harbor of Sinop, Black Sea. Su Urun. Muh. Derg. 2006, 25, 22-27. Das, Y. K. Aksoy, A. Baskaya, R.H. Duyar, A.D. Guvenc, V. Boz,, Heavy metal levels of some marine organisms collected in Samsun and Sinop coasts of Black Sea, in Turkey, Journal of Animal and Veterinary Advances 8(3) (2009)496-499. Nisbet, C.G. Terzi, O. Pilger, N. Sarac, Determination of heavy metal levels in fish samples collected from the Middle Black Sea, Kafkas Univ. Vet. Fak. Derg. 16 (2010)119-125, doi:10.9775/kvfd.2009.982 Ergönül, M.B., Altindağ, A.Heavy metal concentrations in the muscle tissues of seven commercial fish species from Sinop coasts of the Black Sea, Annual Set the Environment Protection, 16 (2014) 34-51. Manthey-Karl, M.; Lehmann, I.; Ostermeyer, U.; Schroder, U. Natural chemical composition of commercial fish species: Characterisation of pangasius, wild and farmed turbot and barramundi. Foods 2016, 5(3), 58, doi:10.3390/foods5030058 Yanchilina A. G., Ryan W. B. F., McManus J.F., Dimitrov P., Dimitrov D., Slavova K, M. Filipova-Marinova, Compilation of geophysical, geochronological, and geochemical evidence indicates a rapid Mediterranean-derived submergence of the Black Sea's shelf and subsequent substantial salinification in the early Holocene, Mar. Geol., 383 (2017)14-34, doi:10.1016/j.margeo.2016.11.001 Black Sea Commission (2007) The Commission of the protection of the Black Sea against pollution, Black Sea Transboundary Diagnostic Analysis. Plavan G., Jitar O., Teodosiu C., Nicoara M., Micu D., Strungaru S.A., Toxic metals in tissues of fishes from the Black Sea and associated human health risk exposure, Environ. Sci. Pollut. Res., 24 (2017) 7776-7787, doi:10.1007/s11356-017-8442-6 Zaitsev Y. P., Alexandrov B. G., Berlinsky N. A., Zenetos A., Europe's biodiversity - biogeographical regions and seas, Seas around Europe The Black Sea - an oxygen-poor sea. European Environment Agency (2002). Jitar O., Teodosiu C., Oros A., Plavan G., Nicoara M., Bioaccumulation of heavy metals in marine organisms from the Romanian sector of the Black Sea, N. Biotechnol., 32 (2014) 369-378, doi:10.1016/j.nbt.2014.11.004 Concalves, C.; Martins, M.; Diniz, M.S.; Costa, M.H.; Cairo, S.; Costa, P.M. May sediment contamination be xenoestrogenic to benthic fish? A case study with Solea senegalensis. Mar Environ Res. 2014, 99, 170-178 Cuevas, N.; Zorita, I.; Costa, P.M.; Larreta, J.; Franco, J. Histopathological baseline levels and confounding factors in common sole (Solea solea) for marine environmental risk assessment. Mar Environ Res. 2015, 110, 162-173, doi:10.1016/j.marenvres.2015.09.002. Samsun, N.; Kalayci, F. Survival rates of black sea turbot (Scophthalmus maeoticus Pallas, 1811) captured by bottom turbot gillnets in different depths and fishing seasons between 1999 and 2004. Turk J Fish Aquat Sci. 2005, 5, 57-62 Whitehead, P.J.P.; Bauchot, M.L.; Hureau, J.C.; Nielsen, J.; Tortonese, E. Fishes of the North-eastern Atlantic and The Mediterranean. UNESCO. 1986, Vol. 3, 1287–1293 https://www.fishbase.se/summary/Scophthalmus-maximus.html Radu G., Radu E., Taxonomically identifier for Black Sea fish species – in romanian language (Determinator al principalelor specii de pesti din Marea Neagra), VIROM Publishing Company, ISBN: 978-973-7895- 33-2, 2008, 504 – 509. Yi Y.J., Zhang S.H., The relationships between fish heavy metal concentrations and fish size in the upper and middle reach of Yangtze River, Procedia Environmental Sciences, 2012, 13, 1699-1707. Merciai R., Guasch H., Kumar A., Sabater S., Trace metal concentration and fish size: Variation among fish species in a Mediterranean river, Ecotoxicology and Environmental Safety, 2014, 107, 154-161. Monsefrad F., Imanpour N., Heidary S., Concentration of heavy and toxic metals Cu, Zn, Cd, Pb and Hg in liver and muscles of Rutilus frisii kutum during spawning season with respect to growth parameters, Iranian Journal of Fisheries Science, 2012, 11 (4), 825-839. Maximov V., Zaharia T., Nicolaev S., Radu G., State of the Romanian Black Sea Turbot (Psetta maxima maoetica) resources. Recherches Marines, 43 (2013) 296-306 Eryilmaz L., Dalyan C., Age, growth, and reproductive biology of turbot, Scophthalmus maximus (Actinopterygii: Pleuronectiformes: Scophthalmidae), from the south-western coasts of Black Sea, Turkey; ACTA ICHTHYOLOGICA ET PISCATORIA (2015) 45 (2): 181–188. Strungaru, S.A.; Nicoara, M.; Jitar, O.; Plavan, G. Influence of urban activity in modifying water parameters, concentration and uptake of heavy metals in Typha latifolia into a river that crosses an industrial city. Journal of Env. Health Sci. & Engineering. 2015, 13:5, 1-15 doi:10.1186/s40201-015-0161-7 Bulska E., Krata A., Kalabun M., Wojciechowski M., On the use of certified reference materials for assuring the quality of results for the determination of mercury in environmental samples, Environmental Science and Pollution Research, 2017, 24 (9), 7889-7897. Jitar, O., Teodosiu C., Oros A., Plavan G., Nicoara M., Bioaccumulation of heavy metals in marine organism from the Romanian sector of the Black Sea, New Biotechnology, 2014, 32, 369-378. Bury N., Walker P.A., Glover C.N., Nutritive metal uptake in teleost fish, The Journal of Experimental Biology, 2003, 206, 11-23. Lall, S.P.; Lewis-McCrea, L.M. Role of nutrients in skeletal metabolism and pathology in fish -An overview. Aquaculture 2007, 267, 3-19, doi:10.1016/j.aquaculture.2007.02.053 FOOD AND AGRICULTURE ORGANIZATION OF THE UNITED NATIONS - UNITED NATIONS DEVELOPMENT PROGRAMME, ADCP/REP/80/11 - Fish Feed Technology, Chapter 7, ISBN 92-5-100901-5, 1980, Wei, Y.; Zhang, J.Y.; Zhang, D.W.; Tu, T.H.; Luo, L.G. Metal concentrations in various fish organs of different fish species from Poyang Lake, China. Ecotoxicol. Environ. Saf. 2014, 104, 182-188, doi:10.1016/j.ecoenv.2014.03.001 Bijvelds, M.J.C.; Velden, V.D.; Kolar, Z.I.; Flik, G. Magnesium transport in freshwater teleosts. J. Exp. Biol. 1998, 201, 1981-1990. Vijayan, D. K.; Jayarani, R.; Singh, D.K.; Chatterjee, N.S.; Mathew, S.; Mohanty, B.P.; Sankar, T.V.; Anandan, R. Comparative studies on nutrient profiling of two deep sea fish (Neoepinnula orientalis and Chlorophthalmus corniger) and brackish water fish (Scatophagus argus). JOBAZ. 2016, 77, 41-48, doi:10.1016/j.jobaz.2016.08.003 Heath, A. Uptake, Accumulation, Biotransformation, and Excretion of Xenobiotics, Water pollution and fish physiology, Library of Congress Cataloging in Publication data, Second Edition, Lewis Publishers, Florida. 1995, 79-86. Simionov I.A., Cristea V., Petrea S.M., Bocioc Sîrbu E., Evaluation of heavy metals concentration dynamics in fish from the Black Sea coastal area: an overview, Environmental Engineering and Management Journal, 2019, 18 (5), 1097-1110. Trifan, A.; Breaban, G.; Sava, D.; Bucur, L.; Toma, C. C.; Miron, A. Heavy metal content in macroalgae from Roumanian Black Sea. Rev. Roum. Chim. 2015, 60, 915-920. El-Moselhy, Kh.M.; Othman, A.I.; El-Azem, H.A.; El-Metwally, M.E.A. Bioaccumulation of heavy metals in some tissues of fish in the Red Sea, Egypt. EJBAS. 2014, 1, 97-105, doi:10.1016/j.ejbas.2014.06.001 Wu, S. M.; Shu, L. H.; Liu, J. H. Anti-oxidative functions of mt2 and smtB mRNA expression in the gills and brain of zebrafish (Danio rerio) upon cadmium exposure. Fish Physiol Biochem. 2016, 42, 1709–1720, DOI 10.1007/s10695-016-0251-1 Yuan, S. S.; Lv, Z. M.; Zhu, A.Y.; Zheng, J. L.; Wu, C. W. Negative effect of chronic cadmium exposure on growth, histology, ultrastructure, antioxidant and innate immune responses in the liver of zebrafish: Preventive role of blue light emitting diodes. Ecotoxicol. Environ. Saf. 2017, 139, 18-26, doi:10.1016/j.ecoenv.2017.01.021

Round 3

Reviewer 2 Report

I thank the authors for providing point-by-point answers to my concerns, and I'm overall satisfied with them. I have only a couple of minor comments to dealt with before formal acceptance:

Line 42 – I think a sentence outlining main conclusions and implications to other contexts, other than the actual, would be useful to increase the interest from potential readers. You may consider for example the sentence from lines 387-398.

Table 2 – please check the zeros on the elements from columns 2 and 3. They range from 0.0001 (Cd) to 10000 (K).

Table 6 – Make the distinction between macro and microelements. I suppose macro are Ca, Mg, Na and K, and micro are Zn, Fe, Cu, Ni and Mn.

Author Response

Distinguished Editorial Border of Journal of Marine Science and Engineering,

To Reviewer # 2,

Line 42 – I think a sentence outlining main conclusions and implications to other contexts, other than the actual, would be useful to increase the interest from potential readers. You may consider for example the sentence from lines 387-398.

Answer: The sentence you have recommended from the Conclusions section was introduced in the abstract (below line 42).

Table 2 – please check the zeros on the elements from columns 2 and 3. They range from 0.0001 (Cd) to 10000 (K).

Answer: The values from the mentioned table, columns 2 and 3, were checked and they are correct. The major variation interval between Cd and K is due to their different abundance in fish tissues. Thus, K concentration is generally expressed in g kg-1, whereas Cd is generally expressed as µg g-1 [see reference below]. However, for this study it was chosen to express the concentration of both elements using the same unit of measurements, respectively µg g-, resulting therefore the aforementioned variation interval.

Lourenco, H.M.; Afonso, C.; Anacleto, P.; Martins, M.F.; Nunes, M.L.; Nino, A.R. Elemental composition of four farmed fish produced in Portugal. Int. J. Food Sci. Nutr., 2012, 63, 853–859, doi:10.3109/09637486.2012.681632

Table 6 – Make the distinction between macro and microelements. I suppose macro are Ca, Mg, Na and K, and micro are Zn, Fe, Cu, Ni and Mn

In this table, the distinction between macro- and micro-elements was made by separating them with a border line, in 2 groups, as mentioned above.

All the changes from this third review report are marked in the manuscript, with blue colour.

On behalf of all authors we would like to thank you for your significant contribution in increasing the quality of the present paper.

Kind regards and best wishes!
